# DiffusionShield: A Watermark for Data Copyright Protection against Generative Diffusion Models

## Abstract

Recently, Generative Diffusion Models (GDMs) have showcased their remarkable capabilities in learning and generating images. A large community of GDMs has naturally emerged, further promoting the diversified applications of GDMs in various fields. However, this unrestricted proliferation has raised serious concerns about copyright protection. For example, artists including painters and photographers are becoming increasingly concerned that GDMs could effortlessly replicate their unique creative works without authorization. In response to these challenges, we introduce a novel watermarking scheme, DiffusionShield, against GDMs. DiffusionShield protects images from copyright infringement through encoding the ownership information into an imperceptible watermark and injecting it into the images. Its watermark can be easily learned by GDMs and will be reproduced in their generated images. By detecting the watermark from generated images, copyright infringement can be exposed with evidence. Benefiting from the uniformity of the watermarks and the joint optimization method, DiffusionShield ensures low distortion of the original image, high watermark detection performance, and the ability to embed lengthy messages. We conduct rigorous and comprehensive experiments to show the effectiveness of DiffusionShield in defending against infringement by GDMs and its superiority over traditional watermarking methods.

## 1 Introduction

Generative diffusion models (GDMs), such as Denoising Diffusion Probabilistic Models (DDPM) Ho et al. (2020) have shown their great potential in generating high-quality images. This has also led to the growth of more advanced techniques, such as DALL·E2 (Ramesh et al., 2022), Stable Diffusion (Rombach et al., 2022), and ControlNet (Zhang & Agrawala, 2023). In general, a GDM learns the distribution of a set of collected images, and can generate images that follow the learned distribution. As these techniques become increasingly popular, concerns have arisen regarding the copyright protection of creative works shared on the Internet. For instance, a fashion company may invest significant resources in designing a new fashion. After the company posts the pictures of this fashion to the public for browsing, an unauthorized entity can train their GDMs to mimic its style and appearance, generating similar images and resulting in products. This infringement highlights the pressing need for copyright protection mechanisms.

To provide protection for creative works, watermark techniques such as Cox et al. (2002); Podilchuk & Delp (2001); Zhu et al. (2018); Navas et al. (2008); Yu et al. (2021) are often applied, which aim to inject (invisible) watermarks into images and then detect them to track the malicious copy and accuse the infringement. However, directly applying these existing methods to GDMs still faces tremendous challenges. Indeed, since existing watermark methods have not specifically been designed for GDMs, they might be hard to learn for GDMs and could disappear in the generated images. Then, the infringement cannot be effectively verified and accused. As empirical evidence in Figure 1, we train two popular GDMs on a CIFAR10 dataset whose samples are watermarked by two representative watermark methods (Navas et al., 2008; Zhu et al., 2018), and we try to detect the watermarks in the GDM-generated images.

The result demonstrates that the watermarks from these methods are either hardly learned and reproduced by GDM (e.g., FRQ (Navas et al., 2008)), or require a very large budget (the extent of image distortion) to partially maintain the watermarks (e.g., HiDDeN (Zhu et al., 2018)). Therefore, dedicated efforts are still greatly desired to developing the watermark technique tailored for GDMs.

In this work, we argue that one critical factor that causes the inefficacy of these existing watermark techniques is the inconsistency of watermark patterns on different data samples. In methods such as (Navas et al., 2008; Zhu et al., 2018), the watermark in each image from one owner is distinct. Thus, GDMs can hardly learn the distribution of watermarks and reproduce them in the generated samples. To address this challenge, we propose **DiffusionShield** which aims

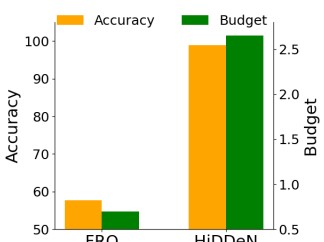

Figure 1: Watermark detection accuracy (%) on GDM-generated images and the corresponding budget ($l_2$ norm) of watermarks.

to enhance the "*pattern uniformity*" (Section 3.2) of the watermarks to make them consistent across different images. We first empirically show that watermarks with pattern uniformity are easy to be reproduced by GDMs in Section 3.2. Then, we provide corresponding theoretic analysis in two examples to demonstrate that the watermarks with pattern uniformity will be learned prior to other features in Section 3.5. The theoretical evidence further suggests that if unauthorized GDMs attempt to learn from the watermarked images, they are likely to learn the watermarks before the original data distribution. To leverage pattern uniformity, DiffusionShield designs a blockwise strategy to divide the watermarks into a sequence of basic patches, and a user has a specific sequence of basic patches which forms a watermark applied on all his/her images and encodes the copyright message. The watermark will repeatedly appear in the training set of GDMs, and thus makes it reproducible and detectable. In the case with multiple users, each user will have his/her own watermark pattern based on encoded message. Furthermore, DiffusionShield introduces a joint optimization method for basic patches and watermark detector to enhance each other, which achieves a smaller budget and higher accuracy. In addition, once the watermarks are obtained, DiffusionShield does not require re-training when there is an influx of new users and images, indicating the flexibility of DiffusionShield to accommodate multiple users. In summary, with the enhanced pattern uniformity in blockwise strategy and the joint optimization, we can successfully secure the data copyright against the infringement by GDMs.

## 2 RELATED WORK

### 2.1 GENERATIVE DIFFUSION MODELS

In recent years, GDMs have made significant strides. A breakthrough in GDMs is achieved by DDPM (Nichol & Dhariwal, 2021), which demonstrates great superiority in generating high-quality images. The work of Ho & Salimans (2022) further advances the field by eliminating the need for classifiers in the training process. Song et al. (2020) presents Denoising Diffusion Implicit Models (DDIMs), a variant of GDMs with improved efficiency in sampling. Besides, techniques such as Rombach et al. (2022) achieve high-resolution image synthesis and text-to-image synthesis. These advancements underscore the growing popularity and efficacy of GDM-based techniques.

To train GDMs, many existing methods rely on collecting a significant amount of training data from public resources (Deng et al., 2009; Yu et al., 2015; Guo et al., 2016). However, there is a concern that if a GDM is trained on copyrighted material and produces outputs similar to the original copyrighted works, it could potentially infringe on the copyright owner's rights. This issue has already garnered public attention (Vincent, 2023), and our paper focuses on mitigating this risk by employing a watermarking technique to detect copyright infringements.

### 2.2 IMAGE WATERMARKING

Image watermarking involves embedding invisible information into the carrier images and is commonly used to identify ownership of the copyright. Traditional watermarking techniques include spatial domain methods and frequency domain methods (Cox et al., 2002; Navas et al., 2008; Shih & Wu, 2003; Kumar, 2020). These techniques embed watermark information by modifying the pixel values (Cox et al., 2002), frequency coefficients (Navas et al., 2008), or both (Shih & Wu, 2003; Kumar, 2020). In recent years, various digital watermarking approaches based on Deep Neural Networks (DNNs) have been proposed. For example, Zhu et al. (2018) uses an autoencoder-based network architecture, while Zhang et al. (2019) designs a GAN for watermrark. Those techniques are then further generalized to photographs (Tancik et al., 2020) and videos (Weng et al., 2019).

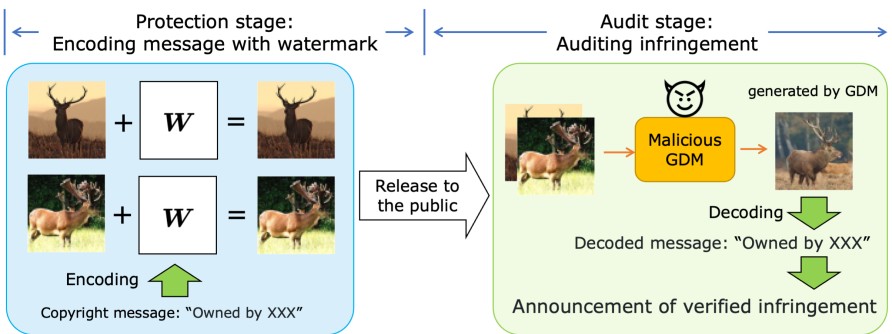

Figure 2: An overview of watermarking with two stages.

Notably, there are existing studies focusing on watermarking generative neural networks, such as GANs (Goodfellow et al., 2020) and image processing networks (Sehwag et al., 2022). Their goal is to safeguard the *intellectual property (IP) of generative models and generated images*, while our method is specifically designed for safeguarding *the copyright of data against potential infringement by these GDMs*. To accomplish their goals, the works (Wu et al., 2020; Yu et al., 2021; Zhao et al., 2023a; Zhang et al., 2020) embed imperceptible watermarks into every output of a generative model, enabling the defender to determine whether an image was generated by a specific model or not. Various approaches have been employed to inject watermarks, including reformulating the training objectives of the generative models (Wu et al., 2020), modifying the model's training data (Yu et al., 2021; Zhao et al., 2023a), or directly applying a watermark embedding process to the output images before they are presented to end-users (Zhang et al., 2020).

## 3 METHOD

In this section, we first formally define the problem and the key notations. Next, we show that the "pattern uniformity" is a key factor for the watermark of generated samples. Based on this, we introduce two essential components of our method, DiffusionShield, i.e., blockwise watermark with pattern uniformity and joint optimization, and then provide theoretic analysis of pattern uniformity.

### 3.1 PROBLEM STATEMENT

In this work, we consider two roles: (1) **a data owner** who holds the copyright of the data, releases them solely for public browsing, and aspires to protect them from being replicated by GDMs, and (2) **a data offender** who employs a GDM on the released data to appropriate the creative works and infringe the copyright. On the other hand, in reality, data are often collected from multiple resources to train GDMs. Thus, we also consider a scenario where there are multiple owners to protect their copyright against GDMs by encoding the copyright information into watermarks. We start by defining the one-owner case, and then extend the discussion to the multiple-owner case:

• **Protection for one-owner case.** An image owner aims to release $n$ images, $\{X_{1:n}\}$, strictly for browsing. Each image $X_i$ has a shape of $(U, V)$ where $U$ and $V$ are the height and width, respectively. As shown in Figure 2, the protection process generally comprises two stages: 1) *a protection stage* when the owner encodes the copyright information into the invisible watermark and adds it to the protected data; and 2) *an audit stage* when the owner examines whether a generated sample infringes upon their data. In the following, we introduce crucial definitions and notations.

1) *The protection stage* happens before the owner releases $\{X_{1:n}\}$ to the public. To protect the copyright, the owner encodes the copyright message $M$ into each of the invisible watermarks $\{W_{1:n}\}$, and adds $W_i$ into $X_i$ to get a protected data $\tilde{X}_i = X_i + W_i$. $M$ can contain information like texts which can signify the owners' unique copyright. The images $\tilde{X}_i$ and $X$ appear similar in human eyes with a small watermark budget $\|W_i\|_p \leq \epsilon$. Instead of releasing $\{X_{1:n}\}$, the owner releases the protected $\{\tilde{X}_{1:n}\}$ for public browsing.

2) *The audit stage* refers to that the owner finds suspicious images which potentially offend the copyright of their images, and they scrutinize whether these images are generated from their released data. We assume that the data offender collects a dataset $\{X_{1:N}^{\mathcal{G}}\}$ that contains the protected images $\{\tilde{X}_{1:n}\}$, i.e. $\{\tilde{X}_{1:n}\} \subset \{X_{1:N}^{\mathcal{G}}\}$ where $N$ is the total number of both protected and unprotected images $(N > n)$, and trains a GDM, $\mathcal{G}$, from scratch to generate images, $X_{\mathcal{G}}$. If $X_{\mathcal{G}}$ contains the copyright information of the data owner, once $X_{\mathcal{G}}$ is inputted to a decoder $\mathcal{D}$, the copyright message should be decoded by $\mathcal{D}$.

• **Protection for multiple-owner case.** When there are $K$ data owners to protect their distinct sets of images, we denote their sets of images as $\{\boldsymbol{X}_{1:n}^k\}$ where $k = 1, ..., K$. Following the methodology of one-owner case, each owner can re-use the same encoding process and decoder to encode and decode distinct messages in different watermarks, $\boldsymbol{W}_i^k$, which signifies their specific copyright messages $\boldsymbol{M}^k$. The protected version of images is denoted by $\tilde{\boldsymbol{X}}_i^k = \boldsymbol{X}_i^k + \boldsymbol{W}_i^k$. Then the protected images, $\{\tilde{\boldsymbol{X}}_{1:n}^k\}$, can be released by their respective owners for public browsing, ensuring their copyright is maintained. More details about the two protection cases can be found in Appendix A.

## 3.2 Pattern Uniformity

In this subsection, we uncover one important factor "*pattern uniformity*" which could be an important reason for the failure of existing watermark techniques. Previous studies (Sehwag et al., 2022; Um & Ye, 2023; Daras et al., 2023) observe that GDMs tend to learn data samples from high probability density regions in the data space and ignore the low probability density regions. However, many existing watermarks such as FRQ (Navas et al., 2008) and HiDDeN (Zhu et al., 2018) can only generate distinct watermarks for different data samples. Since their

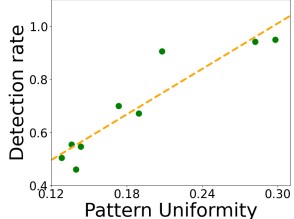

Figure 3: Uniformity vs. watermark detection rate.

generated watermarks are dispersed, these watermarks cannot be effectively extracted and learned. Observing the above, we formally define the "pattern uniformity" as the consistency of different watermarks injected for different samples:

$$Z = 1 - \frac{1}{n}\sum_{i=1}^{n}\left\|\frac{\boldsymbol{W}_i}{\|\boldsymbol{W}_i\|_2} - \boldsymbol{W}_{mean}\right\|_2, \text{ where } \boldsymbol{W}_{mean} = \frac{1}{n}\sum_{i=1}^{n}\frac{\boldsymbol{W}_i}{\|\boldsymbol{W}_i\|_2} \tag{1}$$

where $Z$ corresponds to the standard deviation of normalized watermarks.

We further conduct experiments to illustrate the importance of this "*pattern uniformity*". In the experiment shown in Figure 3, we test DDPM's ability in learning watermarks with different pattern uniformity. The watermarks $\boldsymbol{W}_i$ are random pictures whose pixel value is re-scaled by the budget $\sigma$, and the watermarked images are $\tilde{\boldsymbol{X}}_i = \boldsymbol{X}_i + \sigma \times \boldsymbol{W}_i$. More details about the settings for this watermark and the detector can be found in Appendix C.1. Figure 3 illustrates a positive correlation between the watermark detection rate in the GDM-generated images and the pattern uniformity, which implies that pattern uniformity improves watermark reproduction. Based on pattern uniformity, in Section 3.3 and 3.4, we introduce how to design DiffusionShield, and in Section 3.5, we provide the theoretic analysis of the pattern uniformity based the two examples to justify that the watermarks will be first learned prior to other sparse hidden features and, thus, provide an effective protection.

## 3.3 Watermarks and Decoding Watermarks

In this subsection, we introduce our proposed approach, referred as DiffusionShield. This model is designed to resolve the problem of inadequate reproduction of prior watermarking approaches in generated images. It adopts a blockwise watermarking approach to augment pattern uniformity, which improves the reproduction of watermarks in generated images and enhances flexibility.

**Blockwise watermarks.** In DiffusionShield, to strengthen the pattern uniformity in $\{\boldsymbol{W}_{1:n}\}$, we use the same watermark $\boldsymbol{W}$ for each $\boldsymbol{X}_i$ from the same owner. The sequence of *basic patches* encodes the textual copyright message $\boldsymbol{M}$ of the owner. In detail, $\boldsymbol{M}$ is first converted into a sequence of binary numbers by predefined rules such as ASCII. To condense the sequence's length, we convert the binary sequence into a $B$-nary sequence, denoted as $\{\boldsymbol{b}_{1:m}\}$, where $m$ is the message length and $B$-nary represents different numeral systems like quarternary ($B = 4$) and octal ($B = 8$). Accordingly, DiffusionShield partitions the whole watermark $\boldsymbol{W}$ into a sequence of $m$ patches, $\{\boldsymbol{w}_{1:m}\}$, to represent $\{\boldsymbol{b}_{1:m}\}$. Each patch is chosen from a candidate set of basic patch $\{\boldsymbol{w}^{(1:B)}\}$. The set $\{\boldsymbol{w}^{(1:B)}\}$ has $B$ basic patch candidates with a shape $(u, v)$, which represent different values of the $B$-nary bits. The sequence of $\{\boldsymbol{w}_{1:m}\}$ denotes the $B$-nary bits $\{\boldsymbol{b}_{1:m}\}$ derived from $\boldsymbol{M}$. For example, in Figure 4, we have 4 patches ($B = 4$), and each of the patches has a unique pattern which represents 0, 1, 2, and 3. To encode the copyright message $\boldsymbol{M}$ = "Owned by XXX", we first convert it into binary sequence "01001111 01110111..." based on ASCII, and transfer it into quarternary sequence $\{\boldsymbol{b}_{1:m}\}$, "103313131232...". (The sequence length $m$ should be less or equal to $8 \times 8$, since there are only $8 \times 8$ patches in Figure 4.) Then we concatenate these basic patches in the order of $\{\boldsymbol{b}_{1:m}\}$ for the complete watermark $\boldsymbol{W}$ and add $\boldsymbol{W}$ to each image from the data owner. Once the offender uses GDMs to learn from it, the watermarks will appear in generated images, serving as evidence of infringement.

**Decoding the watermarks.** DiffusionShield employs a decoder $\mathcal{D}_\theta$ by classification in patches, where $\theta$ is the parameters. $\mathcal{D}_\theta$ can classify $\boldsymbol{w}_i$ into a bit $\boldsymbol{b}_i$. The decoder $\mathcal{D}_\theta$ accepts a watermarked image block, $\boldsymbol{x}_i + \boldsymbol{w}_i$, as input and outputs the bit value of $\boldsymbol{w}_i$, i.e., $\boldsymbol{b}_i = \mathcal{D}_\theta(\boldsymbol{x}_i + \boldsymbol{w}_i)$. The suspect generated image is partitioned into a sequence $\{(\boldsymbol{x} + \boldsymbol{w})_{1:m}\}$, and then is classified into $\{\boldsymbol{b}_{1:m}\} = \{\mathcal{D}_\theta(\boldsymbol{x}_i + \boldsymbol{w}_i)|i = 1, ..., m\}$ in a patch-by-patch manner. If $\{\boldsymbol{b}_{1:m}\}$ is the $B$-nary message that we embed into the watermark, we can accurately identify the owner of the data, and reveal the infringement. **Remarks.** Since we assign the same watermark $\boldsymbol{W}$ to each image of one user, the designed watermark evidently has higher uniformity. Additionally, DiffusionShield shows remarkable flexibility when applied to multiple-owner scenarios since basic patches and decoder can be reused by new owners.

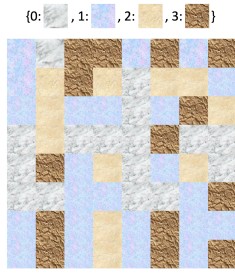

Figure 4: An $8 \times 8$ sequence of basic patches encoded with message "103313131232...". Different patterns represent different basic patches.

### 3.4 JOINTLY OPTIMIZE WATERMARK AND DECODER

While pattern uniformity facilitates the reproduction of watermarks in GDM-generated images, it does not guarantee the detection performance of the decoder, $\mathcal{D}_\theta$. Therefore, we further propose a joint optimization method to search for the optimal basic patch patterns and obtain the optimized detection decoder. Ideally, the basic patches and the decoder should satisfy:

$$\boldsymbol{b}^{(i)} = \mathcal{D}_\theta(\boldsymbol{p} + \boldsymbol{w}^{(i)}) \text{ for } \forall\, i \in \{1, 2, ..., B\}, \tag{2}$$

where $\boldsymbol{w}^{(i)}$ is one of the $B$ basic patch candidates, $\boldsymbol{b}^{(i)}$ is the correct label for $\boldsymbol{w}^{(i)}$, and $\boldsymbol{p}$ can be a random block with the same shape as $\boldsymbol{w}^{(i)}$ cropped from any image. The ideal decoder, capable of accurately predicting all the watermarked blocks, ensures that all embedded information can be decoded from the watermark. To increase the detection performance of the decoder, we simultaneously optimize the basic patches and the decoder using the following bi-level objective:

$$\min_{\boldsymbol{w}^{1:B}} \min_\theta \mathbb{E}\left[\sum_{i=1}^B L_{\mathrm{CE}}\left(\mathcal{D}_\theta\left(\boldsymbol{p} + \boldsymbol{w}^{(i)}\right), \boldsymbol{b}^{(i)}\right)\right] \text{ s.t. } \|\boldsymbol{w}^{(i)}\|_\infty \leq \epsilon, \tag{3}$$

where $L_{\mathrm{CE}}$ is the cross-entropy loss for the classification. The $l_\infty$ budget is constrained by $\epsilon$. To reduce the number of categories of basic patches, we set $\boldsymbol{w}^{(1)} = \boldsymbol{0}$, which means that the blocks without watermark should be classified as $\boldsymbol{b} = 1$. Thus, the bi-level optimization can be rewritten as:

$$\begin{cases} \theta^* = \arg\min_\theta \mathbb{E}\left[\sum_{i=1}^B L_{\mathrm{CE}}\left(\mathcal{D}_\theta\left(\boldsymbol{p} + \boldsymbol{w}^{(i)}\right), \boldsymbol{b}^{(i)}\right)\right] \\ \boldsymbol{w}^{(2:B),*} = \arg\min_{\boldsymbol{w}^{(2:B)}} \mathbb{E}\left[\sum_{i=2}^B L_{\mathrm{CE}}\left(\mathcal{D}_{\theta^*}\left(\boldsymbol{p} + \boldsymbol{w}^{(i)}\right), \boldsymbol{b}^{(i)}\right)\right] \text{ s.t. } \|\boldsymbol{w}^{(i)}\|_\infty \leq \epsilon. \end{cases} \tag{4}$$

The upper-level objective aims to increase the performance of $\mathcal{D}_\theta$, while the lower-level objective optimizes the basic patches to facilitate their detection by the decoder. By the two levels of objectives, the basic patches and decoder potentially promote each other to achieve higher accuracy on smaller budget. To ensure basic patches can be adapted to various image blocks and increase their flexibility, we use randomly cropped image blocks as the host images in the training process of basic patches and decoder. More details about the algorithm of joint optimization can be found in Appendix D.

### 3.5 THEORETIC ANALYSIS OF PATTERN UNIFORMITY BASED ON TWO EXAMPLES

In this subsection, we provide theoretic analysis with two examples, a linear regression model for supervised task, and a multilayer perceptron (MLP) with a general loss function (which can be a **generation** task), to justify that watermarks with pattern uniformity are stronger than other features, and machine learning models can learn features from watermarks earlier and more easily regardless of the type of tasks. Following the same idea, DiffusionShield provides an effective protection since GDMs have to learn watermarks first if they want to learn from protected images.

For both two examples, we use the same assumption for the features in the watermarked dataset. For simplicity, we assume the identical watermark is added onto each sample in the dataset. We impose the following data assumption, which is extended from the existing sparse coding model (Olshausen & Field, 1997; Mairal et al., 2010; Arora et al., 2016; Allen-Zhu & Li, 2022).

**Assumption 1** (Sparse coding model with watermark). *The observed data is $\boldsymbol{Z} = \boldsymbol{MS}$, where $\boldsymbol{M} \in \mathbb{R}^{d \times d}$ is a unitary matrix, and $\boldsymbol{S} = (\boldsymbol{s}_1, \boldsymbol{s}_2, \cdots, \boldsymbol{s}_d)^\top \in \mathbb{R}^d$ is the hidden feature composed of $d$ sparse features:*

$$P(\boldsymbol{s}_i \neq 0) = p, \text{and } \boldsymbol{s}_i^2 = \mathcal{O}(1/pd) \text{ when } \boldsymbol{s}_i \neq 0. \tag{5}$$

*The norm $\|\cdot\|$ is $L_2$ norm. For $\forall i \in [d]$, $\mathbb{E}[\boldsymbol{s}_i] = 0$. The watermarked data is $\tilde{\boldsymbol{Z}} = \boldsymbol{MS} + \boldsymbol{\delta}$, and $\boldsymbol{\delta}$ is a constant watermark vector for all the data samples because of pattern uniformity.*

For the linear regression task, $\boldsymbol{Y} = \boldsymbol{S}^\top \boldsymbol{\beta} + \boldsymbol{\epsilon}$ is the ground truth label, where $\boldsymbol{\epsilon} \sim \mathcal{N}(0, \sigma^2)$ is the noise and $\beta_i = \Theta(1)$ so that $Y^2 = \mathcal{O}_p(1)$. We represent the linear regression model as $\hat{\boldsymbol{Y}} = \tilde{\boldsymbol{Z}}^\top \boldsymbol{w}$, using the watermark data $\tilde{\boldsymbol{Z}}$, where $\boldsymbol{w} \in \mathbb{R}^{1 \times d}$ is the parameter to learn. The mean square error (MSE) loss for linear regression task can be represented as

$$L(\mathbf{w}) = (\tilde{\boldsymbol{Z}}^\top \mathbf{w} - \boldsymbol{S}^\top \boldsymbol{\beta} - \boldsymbol{\epsilon})^2.$$

Given the above problem setup, we have the following result:

**Example 1.** *Consider the initial stage of the training, i.e., $\mathbf{w}$ is initialized with $\mathbf{w}_i \overset{i.i.d.}{\sim} \mathcal{N}(0, 1)$.*

*With Assumption 1, the gradient, with respect to $\mathbf{w}$, of MSE loss for the linear regression model defined above given infinite samples can be derived as*

$$\mathbb{E}\left[\frac{\partial L}{\partial \mathbf{w}}\right] = \mathbb{E}[A(\boldsymbol{S})] + \mathbb{E}[B(\boldsymbol{\delta})], \tag{6}$$

*where $\mathbb{E}[A(\boldsymbol{S})]$ is the hidden feature term that contains the gradient terms from hidden features, and $\mathbb{E}[B(\boldsymbol{\delta})]$ is the watermark term that contains the gradient terms from the watermark.*

*There are three observations. First, watermark is learned prior to other hidden features after initialization. If $\|\boldsymbol{\delta}\| \gg 1/\sqrt{d}$, then with high probability w.r.t. the initialization, $\mathbb{E}\|B(\boldsymbol{\delta})\| \gg \mathbb{E}\|A(\boldsymbol{S})\|$, and $\mathbb{E}\|B(\boldsymbol{\delta})\|$ is maximized with the best uniformity. Second, since $\|\boldsymbol{\delta}\| \ll 1/\sqrt{pd}$, the watermark $\boldsymbol{\delta}$ will be much smaller than any active hidden feature. Finally, when the training converges, the final trained model does not forget $\boldsymbol{\delta}$. (The proof can be found in Appendix B.1.)*

In addition to the linear regression task, we extend our analysis to neural networks with a general loss to further explain the feasibility of the intuition for a generative task. We follow Assumption 1 and give the toy example for neural networks:

**Example 2.** *We use an MLP with $\tilde{\boldsymbol{Z}}$ as input to fit a general loss $L(\boldsymbol{\mathcal{W}}, \tilde{\boldsymbol{Z}})$. $L(\boldsymbol{\mathcal{W}}, \tilde{\boldsymbol{Z}})$ can be a classification or generation task. $\boldsymbol{\mathcal{W}}$ is the parameter of it, and $\boldsymbol{\mathcal{W}}_1$ is the first layer of $\boldsymbol{\mathcal{W}}$. Under mild assumptions, we can derive gradient with respect to each neuron in $\boldsymbol{\mathcal{W}}_1$ into hidden feature term and watermark term as Eq. 6. When $1/\sqrt{d} \ll \|\boldsymbol{\delta}\| \ll 1/\sqrt{pd}$, the watermark term will have more influence and be learned prior to other hidden features in the first layer even though the watermark has a much smaller norm than each active hidden feature. (The proof can be found in Appendix B.2.)*

With the theoretical analysis on the two examples, we justify that the watermark with high pattern uniformity is easier/earlier to be learned than other sparse hidden features. It suggests if the authorized people use GDM to learn from the protected images, the GDM will first learn the watermarks before the data distribution. Therefore, our method can provide an effective protection agaist GDM. We also provide empirical evidence to support this analysis in Appendix B.3.

## 4 EXPERIMENT

In this section, we assess the efficacy of DiffusionShield across various budgets, datasets, and protection scenarios. We first introduce our experimental setups in Section 4.1. In Section 4.2, we evaluate the performance in terms of its accuracy and invisibility. Then we investigate the flexibility and efficacy in multiple-user cases, capacity for message length and robustness, in Section 4.3 to 4.6, respectively. We also evaluate the quality of generated images in Appendix H.

### 4.1 EXPERIMENTAL SETTINGS

**Datasets, baselines and GDM**. We conduct the experiments using four datasets and compare DiffusionShield with four baseline methods. The datasets include CIFAR10 and CIFAR100, both with $(U, V) = (32, 32)$, STL10 with $(U, V) = (64, 64)$ and ImageNet-20 with $(U, V) = (256, 256)$. The baseline methods include Image Blending (IB) which is a simplified version of DiffusionShield without joint optimization, DWT-DCT-SVD based watermarking in the frequency domain (FRQ) (Navas et al., 2008), HiDDeN (Zhu et al., 2018), and DeepFake Fingerprint Detection (DFD) (Yu et al., 2021) (which is designed for DeepFake Detection and adapted to our data protection goal). In the audit

stage, we use the improved DDPM (Nichol & Dhariwal, 2021) as the GDM to train on watermarked data. More details about baselines and improved DDPM is in Appendix C.4 and C.5, respectively.

**Evaluation metrics**. In our experiments, we generate $T$ images from each GDM and decode copyright messages from them. We compare the effectiveness of watermarks in terms of their invisibility, the decoding performance, and the capacity to embed longer messages:

- **(Perturbation) Budget.** We use the LPIPS (Zhang et al., 2018) metric together with $l_2$ and $l_\infty$ differences to measure the visual discrepancies between the original and watermarked images. The lower values of these metrics indicate better invisibility.
- **(Detection) Accuracy.** Following Yu et al. (2021) and Zhao et al. (2023b), we apply bit accuracy to evaluate the correctness of detected messages encoded. To compute bit accuracy, we transform the ground truth $B$-nary message $\{b_{1:m}\}$ and the decoded $\{\hat{b}_{1:m}\}$ back into binary messages $\{b'_{1:m \log_2 B}\}$ and $\{\hat{b}'_{1:m \log_2 B}\}$. The bit accuracy for one watermark is

$$\text{Bit-Acc} \equiv \frac{1}{m \log_2 B} \sum_{k=1}^{m \log_2 B} \mathbb{1}\left(b'_{1:m \log_2 B} = \hat{b}'_{1:m \log_2 B}\right).$$

  The worst bit accuracy is expected to be 50%, which is equivalent to random guessing.
- **Message length.** The length of encoded message reflects the capacity of encoding. To ensure accuracy of FRQ and HiDDeN, we use a 32-bit message for CIFAR images and 64 bits for STL10. For others, we encode 128 bits into CIFAR, 512 bits into STL10 and 256 bits into ImageNet.

**Implementation details**. We set $(u, v) = (4, 4)$ as the shape of the basic patches and set $B = 4$ for quarternary messages. We use ResNet (He et al., 2016) as the decoder to classify different basic patches. For the joint optimization, we use 5-step PGD (Madry et al., 2017) with $l_\infty \leq \epsilon$ to update the basic patches and use SGD to optimize the decoder. As mentioned in Section 3.1, the data offender may collect and train the watermarked images and non-watermarked images together to train GDMs. Hence, in all the datasets, we designate one random class of images as watermarked images, while treating other classes as unprotected images. To generate images of the protected class, we either 1) use a **class-conditional** GDM to generate images from the specified class, or 2) apply a classifier to filter images of the protected class from the **unconditional** GDM's output. The bit accuracy on unconditionally generated images may be lower than that of the conditional generated images since object classifiers cannot achieve 100% accuracy. In the joint optimization, we use SGD with learning rate = 0.01 and weight decay = $5 \times 10^{-4}$ to train the decoder and we use 5-step PGD with step size to be 1/10 of the $L_\infty$ budget to train the basic patches. More details are presented in Appendix C.3.

### 4.2 Results on Protection Performance against GDM

In this subsection, we show that DiffusionShield provides protection with high bit accuracy and good invisibility in Table 1. We compare on two groups of images: (1) the originally released images with watermarks (**Released**) and (2) the generated images from class-conditional GDM or unconditional GDM trained on the watermarked data (**Cond.** and **Uncond.**). Based on Table 1, we can see:

**First**, DiffusionShield can protect the images with the highest bit accuracy and the lowest budget among all the methods. For example, on CIFAR10 and STL10, with all the budgets from 1/255 to 8/255, DiffusionShield can achieve almost 100% bit accuracy on released images and conditionally generated images, which is better than all the baseline methods. Even constrained by the smallest budget with an $l_\infty$ norm of $1/255$, DiffusionShield can still achieve a high successful reproduction rate. On CIFAR100 and ImageNet, DiffusionShield with an $l_\infty$ budget of $4/255$ achieves a higher bit accuracy in generated images with a much lower $l_\infty$ difference and LPIPS than baseline methods. For baselines, FRQ cannot be reproduced by GDM, while HiDDeN and DFD require a much larger perturbation budget over DiffusionShield (Image examples are shown in Appendix E). The accuracy of IB is much worse than the DiffusionShield with 1/255 budget on CIFAR10 and STL10. To explain IB, without joint optimization, the decoder cannot perform well on released images and thus cannot guarantee its accuracy on generated images, indicating the importance of joint optimization.

**Second**, enforcing pattern uniformity can promote the reproduction of watermarks in generated images. In Table 1, we can see that the bit accuracy of the conditionally generated images watermarked by DiffusionShield is as high as that of released images with a proper budget. In addition to DiffusionShield, IB's accuracy in released data and conditionally generated data are also similar. This is because IB is a simplified version of our method without joint optimization and also has high pattern uniformity. In contrast, other methods without pattern uniformity all suffer from a

Table 1: Bit accuracy (%) and budget of the watermark

| | | | IB | FRQ | HiDDeN | DFD | DiffusionShield (ours) | | | |
|---|---|---|---|---|---|---|---|---|---|---|
| CIFAR10 | Budget | $l_\infty$ | 7/255 | 13/255 | 65/255 | 28/255 | **1/255** | 2/255 | 4/255 | 8/255 |
| | | $l_2$ | 0.52 | 0.70 | 2.65 | 1.21 | **0.18** | 0.36 | 0.72 | 1.43 |
| | | LPIPS | 0.01582 | 0.01790 | 0.14924 | 0.07095 | **0.00005** | 0.00020 | 0.00120 | 0.01470 |
| | Accuracy | Released | 87.2767 | 99.7875 | 99.0734 | 95.7763 | 99.6955 | 99.9466 | 99.9909 | **99.9933** |
| | | Cond. | 87.4840 | 57.7469 | 98.9250 | 93.5703 | 99.8992 | 99.9945 | **100.0000** | 99.9996 |
| | | Uncond. | 81.4839 | 55.6907 | **97.1536** | 89.1977 | 93.8186 | 95.0618 | 96.8904 | 96.0877 |
| | Pattern Uniformity | | 0.963 | 0.056 | 0.260 | 0.236 | 0.974 | 0.971 | 0.964 | 0.954 |
| CIFAR100 | Budget | $l_\infty$ | 7/255 | 14/255 | 75/255 | 44/255 | **1/255** | 2/255 | 4/255 | 8/255 |
| | | $l_2$ | 0.52 | 0.69 | 3.80 | 1.58 | **0.18** | 0.36 | 0.72 | 1.43 |
| | | LPIPS | 0.00840 | 0.00641 | 0.16677 | 0.03563 | **0.00009** | 0.00013 | 0.00134 | 0.00672 |
| | Accuracy | Released | 84.6156 | 99.5250 | 99.7000 | 96.1297 | 99.5547 | 99.9297 | 99.9797 | **99.9922** |
| | | Cond. | 54.3406 | 54.4438 | 95.8640 | 90.5828 | 52.0078 | 64.3563 | 99.8000 | **99.9984** |
| | | Uncond. | 52.2786 | 55.5380 | 77.7616 | 77.7961 | 52.8320 | 54.4271 | **91.3021** | 87.2869 |
| | Pattern Uniformity | | 0.822 | 0.107 | 0.161 | 0.180 | 0.854 | 0.855 | 0.836 | 0.816 |
| STL10 | Budget | $l_\infty$ | 8/255 | 14/255 | 119/255 | 36/255 | **1/255** | 2/255 | 4/255 | 8/255 |
| | | $l_2$ | 1.09 | 1.40 | 7.28 | 2.16 | **0.38** | 0.76 | 1.51 | 3.00 |
| | | LPIPS | 0.06947 | 0.02341 | 0.32995 | 0.09174 | **0.00026** | 0.00137 | 0.00817 | 0.03428 |
| | Accuracy | Released | 92.5895 | 99.5750 | 97.2769 | 94.2813 | 99.4969 | 99.9449 | 99.9762 | **99.9926** |
| | | Cond. | 96.0541 | 54.3945 | 96.5164 | 94.7236 | 95.4848 | 99.8164 | 99.8883 | **99.9828** |
| | | Uncond. | 89.2259 | 56.3038 | 91.3919 | 91.8919 | 82.5841 | 93.4693 | **96.1360** | 95.0586 |
| | Pattern Uniformity | | 0.895 | 0.071 | 0.155 | 0.203 | 0.924 | 0.921 | 0.915 | 0.907 |
| ImageNet-20 | Budget | $l_\infty$ | - | 20/255 | 139/255 | 88/255 | **1/255** | 2/255 | 4/255 | 8/255 |
| | | $l_2$ | - | 5.60 | 25.65 | 21.68 | **1.17** | 2.33 | 4.64 | 9.12 |
| | | LPIPS | - | 0.08480 | 0.44775 | 0.30339 | **0.00019** | 0.00125 | 0.00661 | 0.17555 |
| | Accuracy | Released | - | 99.8960 | 98.0625 | 99.3554 | 99.9375 | 99.9970 | 99.9993 | **100.0000** |
| | | Cond. | - | 50.6090 | 98.2500 | 81.3232 | 53.6865 | 53.7597 | 99.9524 | **100.0000** |
| | Pattern Uniformity | | - | 0.061 | 0.033 | 0.041 | 0.941 | 0.930 | 0.908 | 0.885 |

drop of accuracy from released images to conditionally generated images, especially FRQ, which has pattern uniformity lower than 0.11 and an accuracy level on par with a random guess. This implies that the decoded information in watermarks with high pattern uniformity (e.g., IB and ours in CIFAR10 are higher than 0.95) does not change much from released images to generated images and the watermarks can be exactly and easily captured by GDM. Notably, the performance drop on CIFAR100 and ImageNet in 1/255 and 2/255 is also partially due to the low watermark rate. In fact, both a small budget and a low watermark rate can hurt the reproduction of watermarks in generated images. In Appendix F, we discuss the effectiveness when watermark rate is low. We find that in multiple user case, even though the watermark rate for each user is low and they encode different messages and do not share the pattern uniformity, our method can still performs well.

### 4.3 FLEXIBILITY AND EFFICACY IN MULTIPLE-USER CASE

In this subsection, we demonstrate that DiffusionShield is flexible to be transferred to new users while maintaining good protection against GDMs. We assume that multiple copyright owners are using DiffusionShield to protect their images, and different copyright messages should be encoded into the images from different copyright owners. In Table 2, we use one class in the dataset as

Table 2: Average bit accuracy (%) across different numbers of copyright owners (on class-conditional GDM).

| owners | CIFAR-10 | CIFAR-100 |
|---|---|---|
| 1 | 100.0000 | 99.8000 |
| 4 | 99.9986 | 99.9898 |
| 10 | 99.9993 | 99.9986 |

the first owner and the other classes as the new owners. The basic patches (with 4/255 $l_\infty$ budget) and decoder are optimized on the first class and re-used to protect the new classes. Images within the same class have the same message embedded, while images from different classes have distinct messages embedded in them. After reordering the basic patches for different message, transferring from one class to the other classes does not take any additional calculation, and is efficient. We train class-conditional GDM on all of the protected data and get the average bit accuracy across classes. As shown in Table 2, on both CIFAR10 and CIFAR100, when we reorder the basic patches to protect the other 3 classes or 9 classes, the protection performance is almost the same as the one class case, with bit accuracy all close to 100%. Besides flexibility, our watermarks can protect each of the multiple users and can distinguish them clearly even when their data are mixed by the data offender.

### 4.4 GENERALIZATION TO FINE-TUNING GDMs

In this subsection, we test the performance of our method when generalized to the fine-tuning GDMs (Rombach et al., 2022), which is also one of common strategies for learning and generating

Table 3: Bit accuracy (%) in fine-tuning

|  | FRQ | DFD | HiDDeN | Ours |
|---|---|---|---|---|
| $l_2$ | 8.95 | 61.30 | 63.40 | 21.22 |
| Released | 88.86 | 99.20 | 89.48 | 99.50 |
| Generated | 57.13 | 90.31 | 60.16 | 92.88 |

Table 4: Bit accuracy (%) under corruptions

|  | DFD | HiDDeN | Ours |
|---|---|---|---|
| No corrupt | 93.57 | 98.93 | 99.99 |
| Gaussian noise | 68.63 | 83.59 | 81.93 |
| Low-pass filter | 88.94 | 81.05 | 99.86 |
| Greyscale | 50.82 | 97.81 | 99.81 |
| JPEG comp. | 62.52 | 74.84 | 94.45 |

images. Fine-tuning is a more difficult task compared the training-from-scratch setting, because fine-tuning only changes the GDM parameters in a limited extent. This change may be not sufficient to learn all the features in the fine-tuned dataset, therefore, the priority by pattern uniformity becomes even more important. To better generalize our method to the fine-tuning case, we enhance the uniformity in hidden space instead of the pixel space, and limit $l_2$ norm instead of $l_\infty$ norm. More details of fine-tuning and its experiment settings can be found in Appendix I. We assume that the data offender fine-tunes Stable Diffusion (Rombach et al., 2022) to learn the style of *pokemon-blip-captions* dataset (Pinkney, 2022). In Table 3, we compare the budget and bit accuracy of our method with three baselines. The observation is similar to that in Table 1. Although FRQ has smaller budget than ours, the bit accuracy on generated images are much worse. DFD has bit accuracy of 90.31%, but the budget is three times of ours. HiDDeN is worse than ours in both budget and bit accuracy. In summary, our method has the highest accuracy in both released data and generated data.

## 4.5 Capacity for Message Length

The capacity of embedding longer messages is important for watermarking methods since encoding more information can provide more conclusive evidence of infringement. In this subsection, we show the superiority of DiffusionShield over other methods in achieving high watermark capacity. Figure 5 shows the bit accuracy and $l_2$ budgets of watermarks with different message

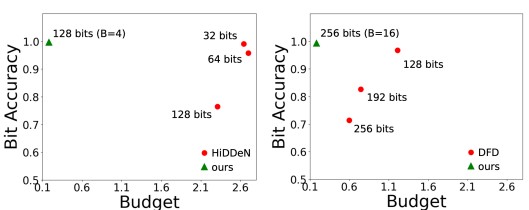

(a) HiDDeN & ours (1/255)  (b) DFD & ours (1/255)
Figure 5: Bit acc. and $l_2$ of different message lengths

lengths on the released protected images in CIFAR10. In Figure 5a, we can see that HiDDeN consistently requires a large budget across varying message lengths, and its accuracy diminishes to 77% at 128 bits. Conversely, DiffusionShield maintains nearly 100% accuracy at 128 bits, even with a much smaller budget. Similarly, in Figure 5b, ours maintains longer capacity with better accuracy and budget than DFD. This indicate that DiffusionShield has much greater capacity compared to HiDDeN and DFD and can maintain good performance even with increased message lengths.

## 4.6 Robustness of DiffusionShield

Robustness of watermarks is important since there is a risk that the watermarks may be distorted by disturbances, such as image corruption due to deliberate post-processing activities during the images' circulation, the application of speeding-up sampling methods in the GDM (Song et al., 2020), or different training hyper-parameters used to train GDM. This subsection demonstrate that DiffusionShield is robust in accuracy on generated images when corrupted. In Appendix G.1 and G.2, we show similar conclusions when sampling procedure is fastened and hyper-parameters are changed.

We consider Gaussian noise, low-pass filter, greyscale and JPEG compression to test the robustness of DiffusionShield against image corruptions. Different from the previous experiments, during the protection stage, we augment our method by incorporating corruptions into the joint optimization. Each corruption is employed after the basic patches are added to the images. Table 4 shows the bit accuracy of DiffusionShield (with an $l_\infty$ budget of 8/255) on corrupted generated images. It maintains around 99.8% accuracy under Greyscale and low-pass filter, nearly matching the accuracy achieved without any corruption. In other corruptions, our method performs better than baselines except HiDDeN in Gaussian noise. In contrast, DFD has a significant reduce in Gaussian noise, Greyscale and JPEG compression, and HiDDeN shows a poor performance under low-pass filter and JPEG Compression. From these results, we can see that DiffusionShield is robust against image corruptions.

## 5 Conclusion

In this paper, we introduce DiffusionShield, a watermark to protect data copyright, which is motivated by our observation that the pattern uniformity can effectively assist the watermark to be captured by GDMs. By enhancing the pattern uniformity of watermarks and leveraging a joint optimization method, DiffusionShield successfully secures copyright with better accuracy and a smaller budget. Theoretic analysis and experimental results demonstrate the superior performance of DiffusionShield.

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

## A    WATERMARKING PROTECTION FOR MULTIPLE COPYRIGHT OWNERS

As shown in Algorithm 1, to extend the protection from the one-owner case to the multiple-owner case, we first build the watermark protection for one owner and get the corresponding watermark decoder $\mathcal{D}_\theta$ (line 1). Then we use the same procedure (that can be decoded by $\mathcal{D}_\theta$) to watermark all the images from other owners (lines 2 to 4).

---

**Algorithm 1** Watermark protection for multiple copyright owners

---

**Input:** The number of distinct sets of images to protect, $K$. Distinct sets, $\{\boldsymbol{X}_{1:n}^k\}$ and different copyright messages for different owners $\boldsymbol{M}^k$, where $k = 1, 2, 3, ..., K$.
**Output:** Watermarked images $\{\tilde{\boldsymbol{X}}_{1:n}^k\}$, where $k = 1, 2, 3, ..., K$ and the watermark decoder $\mathcal{D}_\theta$.
  1: $\{\tilde{\boldsymbol{X}}_{1:n}^1\}, \mathcal{D}_\theta \leftarrow OneOwnerCaseProtection(\{\boldsymbol{X}_{1:n}^1\}, \boldsymbol{M}^1)$
  2: **for** $k = 2$ to K **do**
  3:     $\{\tilde{\boldsymbol{X}}_{1:n}^k\} \leftarrow ReuseEncodingProcess(\{\boldsymbol{X}_{1:n}^k\}, \boldsymbol{M}^k)$
  4: **end for**
  5: return $\{\tilde{\boldsymbol{X}}_{1:n}^k\}$, $k = 1, 2, 3, ..., K$ and $\mathcal{D}_\theta$.

---

## B    THEORETIC ANALYSIS ON TWO EXAMPLES

In this section, we use two examples, linear regression model and MLP, to show that watermarks with high pattern uniformity can be a stronger feature than others and can be learned easier/earlier than other features. We use MSE as the loss of linear regression and use a general loss in MLP to discuss a general case. We provide the theoretical examples in the two examples to explain that the watermarks with pattern uniformity can be learned prior to other features in the optimization starting at the initialized model.

### B.1    LINEAR REGRESSION

*Proof of Example 1.* To reduce the loss by gradient descent, we derive the gradient of $L$ with respect to $\boldsymbol{w}$:

$$
\begin{aligned}
\mathbb{E}\left[\frac{\partial L}{\partial \mathbf{w}}\right] &= \mathbb{E}\left[\frac{\partial(\tilde{\boldsymbol{Z}}^\top \mathbf{w} - \boldsymbol{S}^\top \boldsymbol{\beta} - \boldsymbol{\epsilon})^2}{\partial \mathbf{w}}\right] \\
&= 2\mathbb{E}\left[\tilde{\boldsymbol{Z}}(\tilde{\boldsymbol{Z}}^\top \mathbf{w} - \boldsymbol{S}^\top \boldsymbol{\beta} - \boldsymbol{\epsilon})\right] \\
&= 2\mathbb{E}\left[\tilde{\boldsymbol{Z}}(\tilde{\boldsymbol{Z}}^\top \mathbf{w})\right] - 2\mathbb{E}\left[\tilde{\boldsymbol{Z}}(\boldsymbol{S}^\top \boldsymbol{\beta} + \boldsymbol{\epsilon})\right] \\
&= 2\mathbb{E}\left[(\boldsymbol{MS} + \boldsymbol{\delta})(\boldsymbol{MS} + \boldsymbol{\delta})^\top \mathbf{w}\right] - 2\mathbb{E}\left[(\boldsymbol{MS} + \boldsymbol{\delta})(\boldsymbol{S}^\top \boldsymbol{\beta} + \boldsymbol{\epsilon})\right] \\
&= 2(\mathbb{E}\left[\boldsymbol{MSS}^\top \boldsymbol{M}^\top\right] + \mathbb{E}\left[\boldsymbol{\delta\delta}^\top\right])\mathbf{w} - 2(\mathbb{E}\left[\boldsymbol{MSS}^\top \boldsymbol{\beta}\right] + \mathbb{E}\left[\boldsymbol{\delta S}^\top \boldsymbol{\beta}\right]) - 2\mathbb{E}\left[(\boldsymbol{MS} + \boldsymbol{\delta})\boldsymbol{\epsilon}\right] \\
&= 2(\mathbb{E}\left[\boldsymbol{MSS}^\top \boldsymbol{M}^\top\right] + \mathbb{E}\left[\boldsymbol{\delta\delta}^\top\right])\mathbf{w} - 2\mathbb{E}\left[\boldsymbol{MSS}^\top \boldsymbol{\beta}\right].
\end{aligned}
\tag{7}
$$

In the above gradient, we separate the hidden feature term according to whether it contains $\mathbf{w}$ to make the comparison with terms with and without $\mathbf{w}$ in watermark term.

For $(\mathbb{E}\left[\boldsymbol{MSS}^\top \boldsymbol{M}^\top\right] + \mathbb{E}\left[\boldsymbol{\delta\delta}^\top\right])\mathbf{w}$, we transform the gradient by $\boldsymbol{M}^\top$ to compare the influence on $\boldsymbol{S}$ by each dimension $\boldsymbol{s}_i$. The norm of the two terms are

$$
\begin{aligned}
\left(\boldsymbol{M}^\top \mathbb{E}\left[\boldsymbol{MSS}^\top \boldsymbol{M}^\top\right]\mathbf{w}\right)_i &= \left(\mathbb{E}\left[\boldsymbol{SS}^\top\right]\boldsymbol{M}^\top \mathbf{w}\right)_i \\
&= \mathcal{O}\left(\frac{1}{d}\left\|\boldsymbol{M}_i^\top \mathbf{w}\right\|\right) \\
&= \mathcal{O}_p\left(\frac{1}{d}\right),
\end{aligned}
\tag{8}
$$

and

$$\left\| \boldsymbol{M}^\top \mathbb{E}\left[ \boldsymbol{\delta}\boldsymbol{\delta}^\top \mathbf{w} \right] \right\| = \left\| \mathbb{E}\left[ \boldsymbol{\delta}\boldsymbol{\delta}^\top \mathbf{w} \right] \right\| = \left\| \mathbb{E}\left[ \boldsymbol{\delta} \right] \right\| \times \mathcal{O}\left( \|\boldsymbol{\delta}\| \right) = \mathcal{O}\left( \|\boldsymbol{\delta}\|^2 \right). \tag{9}$$

When $\|\boldsymbol{\delta}\| \gg 1/\sqrt{d}$, the norm of the watermark term in Eq. 9 is larger than the gradient term from each hidden feature in Eq. 8, which means the watermark feature is learned prior to other hidden features in the first optimization step after model is random initialized.

Similarly, for the rest part in the gradient of Eq. 7, we have

$$\left( \boldsymbol{M}^\top \mathbb{E}\left[ \boldsymbol{M}\boldsymbol{S}\boldsymbol{S}^\top \boldsymbol{\beta} \right] \right)_i = \mathcal{O}\left( \frac{1}{d} \left( \boldsymbol{I}_d \boldsymbol{\beta} \right)_i \right) = \mathcal{O}\left( \frac{1}{d}\beta_i \right) = \mathcal{O}\left( \frac{1}{d} \right). \tag{10}$$

When $\|\boldsymbol{\delta}\| \gg 1/\sqrt{d}$, the watermark term in Eq. 9 will have a larger norm than Eq. 10 and the watermark feature can be learned prior to other features.

Combining the other side, when $1/\sqrt{d} \ll \|\boldsymbol{\delta}\| \ll 1/\sqrt{pd}$, because of pattern uniformity, the watermark will have more influence and be learned prior to other hidden features after random initialization even though the watermark has a much smaller norm than each active hidden feature.

On the other hand, assume the watermark $\boldsymbol{\delta}$ has a worse pattern uniformity, and $\boldsymbol{\delta}$ is independent with $\boldsymbol{Z}$. Then the sum of all eigenvalues $\lambda_i(\mathbb{E}[\boldsymbol{\delta}\boldsymbol{\delta}^\top])$ is unchanged, i.e.,

$$\sum_i \lambda_i(\mathbb{E}[\boldsymbol{\delta}\boldsymbol{\delta}^\top]) = tr\left( \mathbb{E}[\boldsymbol{\delta}\boldsymbol{\delta}^\top] \right) = \mathbb{E}tr[\boldsymbol{\delta}\boldsymbol{\delta}^\top] = \mathbb{E}\|\boldsymbol{\delta}\|^2.$$

However, since $\boldsymbol{\delta}$ is random, there are more $\lambda_i$s which are not zero. Consequently, if we look at the $\left\| \mathbb{E}\left[ \boldsymbol{\delta}\boldsymbol{\delta}^\top \mathbf{w} \right] \right\|$, we study

$$\mathbb{E}_{\mathbf{w}} \left\| \mathbb{E}_{\boldsymbol{\delta}}\left[ \boldsymbol{\delta}\boldsymbol{\delta}^\top \mathbf{w} \right] \right\|^2 = tr\left( \mathbb{E}[\boldsymbol{\delta}\boldsymbol{\delta}^\top]\mathbb{E}[\boldsymbol{\delta}\boldsymbol{\delta}^\top] \right) = \sum_i \lambda_i(\mathbb{E}[\boldsymbol{\delta}\boldsymbol{\delta}^\top])^2,$$

and then we can find that the average $\left\| \mathbb{E}\left[ \boldsymbol{\delta}\boldsymbol{\delta}^\top \mathbf{w} \right] \right\|$ becomes smaller.

On the other hand, it is also easy to figure out that the best $\mathbf{w}$ to minimize $L$ is

$$\mathbf{w}^* = (\boldsymbol{I}_d + \mathbb{E}\boldsymbol{\epsilon}\boldsymbol{\epsilon}^\top + \boldsymbol{\delta}\boldsymbol{\delta}^\top)^{-1}\boldsymbol{M}\boldsymbol{\beta},$$

i.e., the training process does not forget $\boldsymbol{\delta}$ in the end. $\qquad\square$

## B.2 Neural Network with a General Task

**Remark 1.** *While one can obtain a closed-form solution in Example 1 for linear regression problem, in Example 2, there is no closed-form solution of the trained neural network. Although theoretically tracking the behavior of the neural network is beyond our scope, we highlight that in existing theoretical studies, e.g., Ba et al. (2019); Allen-Zhu & Li (2022), the neural network will not forget any learned features during the training.*

*Proof of Example 2.* We denote one of the neurons in $\boldsymbol{\mathcal{W}}_1$ as $\mathbf{w}_h$ and shorten the notation of $L(\boldsymbol{\mathcal{W}}, \boldsymbol{S})$ as $L$. In the following, we proof that the gradient updating of each neuron in the first layer is dominated by the $\boldsymbol{\delta}$ because the watermark term has a larger norm compared with other hidden features.

We first derive the gradient of $L$ with respect to $\mathbf{w}_h$:

$$\frac{\partial L}{\partial \mathbf{w}_h} = \frac{\partial L}{\partial \mathbf{w}_h^\top \tilde{\boldsymbol{Z}}} \frac{\partial \mathbf{w}_h^\top \tilde{\boldsymbol{Z}}}{\partial \mathbf{w}_h} = \frac{\partial L}{\partial \mathbf{w}_h^\top \tilde{\boldsymbol{Z}}} \tilde{\boldsymbol{Z}} = \frac{\partial L}{\partial \mathbf{w}_h^\top \tilde{\boldsymbol{Z}}} \left( \boldsymbol{M}\boldsymbol{S} + \boldsymbol{\delta} \right).$$

By denoting $\frac{\partial L}{\partial \mathbf{w}_h^\top \tilde{\boldsymbol{Z}}}$ as $\rho(\tilde{\boldsymbol{Z}})$, we get

$$\frac{\partial L}{\partial \mathbf{w}_h} = \rho(\tilde{\boldsymbol{Z}}) \left( \boldsymbol{M}\boldsymbol{S} + \boldsymbol{\delta} \right).$$

For simplicity, we assume $\boldsymbol{M} = \boldsymbol{I}_d$. Then the gradient is

$$\frac{\partial L}{\partial \mathbf{w}_h} = \rho(\tilde{\boldsymbol{Z}})\left(\boldsymbol{S} + \boldsymbol{\delta}\right).$$

To compare the norm of gradient related to $\boldsymbol{x}_i$ with watermark term, we define $\boldsymbol{S}_{-i} = (\boldsymbol{s}_1, ..., \boldsymbol{s}_{i-1}, 0, \boldsymbol{s}_{i+1}, ..., \boldsymbol{s}_d)$, and $\boldsymbol{S}_i = (0, ..., 0, \boldsymbol{s}_i, 0, ..., 0)$. Then

$$
\begin{aligned}
\frac{\partial L}{\partial \mathbf{w}_h} &= \rho(\tilde{\boldsymbol{Z}})\left(\boldsymbol{S}_{-i} + \boldsymbol{S}_i + \boldsymbol{\delta}\right) \\
&= \rho(\boldsymbol{S}_{-i} + \boldsymbol{S}_i + \boldsymbol{\delta})\left(\boldsymbol{S}_{-i} + \boldsymbol{S}_i + \boldsymbol{\delta}\right) \\
&= \left[\rho\left(\boldsymbol{S}_{-i}\right) + \rho'\left(\boldsymbol{S}_{-i}\right)^\top \left(\boldsymbol{S}_i + \boldsymbol{\delta}\right) + \frac{1}{2}\|\boldsymbol{S}_i + \boldsymbol{\delta}\|^2_{\rho''(\boldsymbol{S}_{-i})} + \mathcal{O}\left(\|\boldsymbol{S}_i + \boldsymbol{\delta}\|^3\right)\right]\left(\boldsymbol{S}_{-i} + \boldsymbol{S}_i + \boldsymbol{\delta}\right) \\
&= \rho\left(\boldsymbol{S}_{-i}\right)\left(\boldsymbol{S}_{-i} + \boldsymbol{S}_i + \boldsymbol{\delta}\right) + \rho'\left(\boldsymbol{S}_{-i}\right)^\top \left(\boldsymbol{S}_i + \boldsymbol{\delta}\right)\left(\boldsymbol{S}_{-i} + \boldsymbol{S}_i + \boldsymbol{\delta}\right) \\
&\quad + \frac{1}{2}\|\boldsymbol{S}_i + \boldsymbol{\delta}\|^2_{\rho''(\boldsymbol{S}_{-i})}\boldsymbol{S}_{-i} + \mathcal{O}\left(\|\boldsymbol{S}_i + \boldsymbol{\delta}\|^3\right) \\
&= \rho(\boldsymbol{S}_{-i})\boldsymbol{S}_{-i} + \rho'(\boldsymbol{S}_{-i})^\top(\boldsymbol{S}_i + \boldsymbol{\delta})\boldsymbol{S}_{-i} \\
&\quad + \rho(\boldsymbol{S}_{-i})(\boldsymbol{S}_i + \boldsymbol{\delta}) + \rho'(\boldsymbol{S}_{-i})^\top(\boldsymbol{S}_i + \boldsymbol{\delta})(\boldsymbol{S}_i + \boldsymbol{\delta}) + \frac{1}{2}\|\boldsymbol{S}_i + \boldsymbol{\delta}\|^2_{\rho''(\boldsymbol{S}_{-i})}\boldsymbol{S}_{-i} \\
&\quad + \mathcal{O}\left(\|\boldsymbol{S}_i + \boldsymbol{\delta}\|^3\right).
\end{aligned}
$$

We further assume $\mathbb{E}\rho(\boldsymbol{S}_{-i}) = 0$, $\mathbb{E}\rho'(\boldsymbol{S}_{-i})\boldsymbol{S}_{-i}^\top = 0$, $\mathbb{E}\rho'(\boldsymbol{S}_{-i})^\top\boldsymbol{\delta} = \Theta(\|\boldsymbol{\delta}\|\|\mathbb{E}\rho'(\boldsymbol{S}_{-i})\|)$, and $\|\mathbb{E}\|\boldsymbol{a}\|^2_{\rho''(\boldsymbol{S}_{-i})}\boldsymbol{S}_{-i}\| = \Theta(\|\boldsymbol{a}\|\|\mathbb{E}\rho'(\boldsymbol{S}_{-i})\|)$ for any proper vector $\boldsymbol{a}$[1]. Taking the expectation of the gradient,

$$
\begin{aligned}
\mathbb{E}_{\boldsymbol{S}}\left[\frac{\partial L}{\partial \mathbf{w}_h}\right] &= \underbrace{\mathbb{E}\rho(\boldsymbol{S}_{-i})\boldsymbol{S}_{-i}}_{=0} + \underbrace{\mathbb{E}\rho'(\boldsymbol{S}_{-i})^\top\boldsymbol{S}_{-i}(\boldsymbol{S}_i + \boldsymbol{\delta})}_{=0} \\
&\quad + \underbrace{\mathbb{E}\rho(\boldsymbol{S}_{-i})(\boldsymbol{S}_i + \boldsymbol{\delta})}_{=0} + \mathbb{E}\rho'(\boldsymbol{S}_{-i})^\top(\boldsymbol{S}_i + \boldsymbol{\delta})(\boldsymbol{S}_i + \boldsymbol{\delta}) + \mathbb{E}\frac{1}{2}\|\boldsymbol{S}_i + \boldsymbol{\delta}\|^2_{\rho''(\boldsymbol{S}_{-i})}\boldsymbol{S}_{-i} \\
&\quad + \underbrace{\mathcal{O}\left(\|\boldsymbol{S}_i + \boldsymbol{\delta}\|^3\right)}_{\text{negligible}} \\
&= \mathbb{E}(\boldsymbol{S}_i + \boldsymbol{\delta})(\boldsymbol{S}_i + \boldsymbol{\delta})^\top\mathbb{E}\rho'(\boldsymbol{S}_{-i}) + \mathbb{E}\frac{1}{2}\|\boldsymbol{S}_i + \boldsymbol{\delta}\|^2_{\rho''(\boldsymbol{S}_{-i})}\boldsymbol{S}_{-i} + o \\
&= \left(\mathbb{E}\boldsymbol{S}_i\boldsymbol{S}_i^\top + \boldsymbol{\delta}\boldsymbol{\delta}^\top\right)\mathbb{E}\rho'(\boldsymbol{S}_{-i}) + \frac{1}{2}\mathbb{E}_{\boldsymbol{S}_{-i}}\left(\mathbb{E}_{\boldsymbol{S}_i}\|\boldsymbol{S}_i\|^2_{\rho''(\boldsymbol{S}_{-i})} + \|\boldsymbol{\delta}\|^2_{\rho''(\boldsymbol{S}_{-i})}\right)\boldsymbol{S}_{-i} + o.
\end{aligned}
$$

The notation $o$ represents negligible terms.

Since $\mathbb{E}\rho'(\boldsymbol{S}_{-i})^\top\boldsymbol{\delta} = \Theta(\|\boldsymbol{\delta}\|\|\mathbb{E}\rho'(\boldsymbol{S}_{-i})\|)$, when $\|\boldsymbol{\delta}\| \gg \mathbb{E}[\boldsymbol{S}_i]$, we have

$$\left\|\left(\mathbb{E}\boldsymbol{S}_i\boldsymbol{S}_i^\top\right)\mathbb{E}\rho'(\boldsymbol{S}_{-i})\right\| \ll \left\|\left(\boldsymbol{\delta}\boldsymbol{\delta}^\top\right)\mathbb{E}\rho'(\boldsymbol{S}_{-i})\right\|.$$

On the other hand, since $\|\mathbb{E}\|\boldsymbol{a}\|^2_{\rho''(\boldsymbol{S}_{-i})}\boldsymbol{S}_{-i}\| = \Theta(\|\boldsymbol{a}\|\|\mathbb{E}\rho'(\boldsymbol{S}_{-i})\|)$, when $\|\boldsymbol{\delta}\| \gg \mathbb{E}[\boldsymbol{S}_i]$, we have

$$\left\|\mathbb{E}_{\boldsymbol{S}_{-i}}\left(\mathbb{E}_{\boldsymbol{S}_{-i}}\|\boldsymbol{S}_i\|^2_{\rho''(\boldsymbol{S}_{-i})}\right)\boldsymbol{S}_{-i}\right\| \ll \left\|\mathbb{E}_{\boldsymbol{S}_{-i}}\left(\|\boldsymbol{\delta}\|^2_{\rho''(\boldsymbol{S}_{-i})}\right)\boldsymbol{S}_{-i}\right\|.$$

To summarize, in general, when $\|\boldsymbol{\delta}\| \gg \mathbb{E}[\boldsymbol{S}_i]$, i.e. $\|\boldsymbol{\delta}\| \gg 1/\sqrt{d}$, the norm of the watermark term in the gradient will be much larger than than expectation of any hidden feature, which means the watermark will be learned prior to other features.

The effect of uniformity of $\boldsymbol{\delta}$ follows the same as in Example 1.

$\square$

---

[1]To simplify the analysis, we directly connect $\|\mathbb{E}\|\boldsymbol{a}\|^2_{\rho''(\boldsymbol{S}_{-i})}\boldsymbol{S}_{-i}\|$ to $\|\boldsymbol{a}\|$. To relax this condition, one may consider imposing proper assumptions to exactly derive the formula of $\|\mathbb{E}\|\boldsymbol{a}\|^2_{\rho''(\boldsymbol{S}_{-i})}\boldsymbol{S}_{-i}\|$. We also avoid extreme cases where terms cancel with each other, e.g., $\boldsymbol{\delta}\boldsymbol{\delta}^\top\mathbb{E}\rho'(\boldsymbol{S}_{-i}) = -\mathbb{E}_{\boldsymbol{S}}\|\boldsymbol{\delta}\|^2_{\rho''(\boldsymbol{S}_{-i})}\boldsymbol{S}_{-i}/2$

### B.3 Experiment to Support Theoretic Analysis with the Two Examples

We use DDPM to learn a watermarked *bird* class in CIFAR10 and compare the accuracy and the quality of generated images in different steps of the training process. The results in Figure 6 show that watermark is much earlier learned before the semantic features, which is consistent with our theoretic analysis in the two examples. In Figure 6, we can see that, at step 20k, the watermark accuracy in generated images is already 94%, but the generated image has no visible feature of bird at all. The bird is generated in high quality until step 60k. This means the watermark is learned much earlier than the semantic features of the images. The observation aligns with our theoretic analysis.

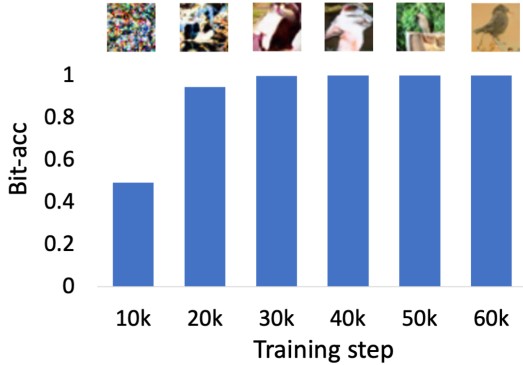

Figure 6: The change of bit accuracy and generated images in the training process.

## C Additional Details of Experimental Settings

### C.1 Watermarks and Detector of Experiment for Pattern Uniformity in Section 3.2

In the experiment shown in Figure 3, we test the ability of DDPM Ho et al. (2020) to learn watermarks with different pattern uniformity and show more details about the setting in this subsection.

**Watermarks.** We first choose one class from CIFAR10 as images requiring watermarks $X_{1:R}$, where $R$ is the number of images in this class and $R = 5000$ for CIFAR10. We randomly choose $C$ images from 5 classes from CIFAR10 as $W_{1:R}$, where $C$ is the number of different watermarks and $C = 5, 10, 15, ....$ Different watermarks are repeatedly added into $X_{1:R}$ by $\tilde{X}_i = X_i + \sigma \times W_i$. For example, we choose $C = 10$ images as watermarks and every watermark is used to watermark $R/C = 500$ images in $X_{1:R}$. By choosing different $C$, we can control the uniformity. Larger $C$ means more diverse watermarks and thus smaller pattern uniformity.

**Detector.** We train a classifier as the detector to detect the watermark in the generated images. The classifier is trained on the images watermarked by 10 classes. The label of the training images is set to be the watermark class. If the classifier predicts that the GDM-generated images have the watermark within the 5 classes from which the $C$ watermarks are chosen, we see it as a successful detection, otherwise it is unsuccessful.

### C.2 Block size and message length for different datasets

In our experiment, we considered four datasets, including CIFAR10 and CIFAR100, both with $(U, V) = (32, 32)$, STL10 with $(U, V) = (64, 64)$ and ImageNet-20 with $(U, V) = (256, 256)$. For CIFAR10, CIFAR100 and STL10, we consider the block size (u, v) = (4,4) and B = 4. For ImageNet-20, we set (u,v) = (16, 16) and B = 2. Therefore, for CIFAR10 and CIFAR100, we are able to encode $\left(\frac{32}{4}\right) \times \left(\frac{32}{4}\right) \times 2 = 128$ bit. For STL-10, we can embed $\left(\frac{64}{4}\right) \times \left(\frac{64}{4}\right) \times 2 = 512$ bit. And for ImageNet, the message length is $\left(\frac{256}{16}\right) \times \left(\frac{256}{16}\right) = 256$ bits.

## C.3 Decoder Architecture and Details about Training Parameters.

Given the small size of the blocks ($4 \times 4$), we adapt the original ResNet structure by including only two residual blocks with 64 filters each, positioned between the initial convolutional layer and the global average pooling layer. In the joint optimization, for training decoder, we use the SGD optimizer with momentum to be 0.9, learning rate to be 0.01 and weight decay to be $5 \times 10^{-4}$, while for training watermark basic patches, we use 5-step PGD with step size to be 1/10 of the $L_\infty$ budget.

## C.4 Details of Baselines

Our method is compared with four existing watermarking methods although they are not specifically designed for the protection of image copyright against GDMs. Information on the baseline methods is provided as follows:

- **Image Blending (IB)**, a simplified version of our approach, which also applies blockwise watermark to achieve pattern uniformity but the patches are not optimized. Instead, it randomly selects some natural images, re-scales their pixel values to 8/255, and uses these as the basic patches. A trained classifier is also required to distinguish which patch is added to a block.

- **DWT-DCT-SVD based watermarking (FRQ)**, one of the traditional watermarking schemes based on the frequency domains of images. It uses Discrete Wavelet Transform (DWT) to decompose the image into different frequency bands, Discrete Cosine Transform (DCT) to separate the high-frequency and low-frequency components of the image, and Singular Value Decomposition (SVD) to embed the watermark by modifying the singular values of the DCT coefficients.

- **HiDDeN** Zhu et al. (2018), a neural network-based framework for data hiding in images. The model comprises a network architecture that includes an encoding network to hide information in an image, a decoding network to extract the hidden information from the image, and a noise network to attack the system, making the watermark robust. In our main experiments, we did not incorporate noise layers into HiDDeN, except during tests of its robustness to noise (Experiments in 4.6).

- **DeepFake Fingerprint Detection (DFD) Yu et al. (2021)**, a method for Deepfake detection and attribution (trace the model source that generated a deepfake). The fingerprint is developed as a unique pattern or signature that a generative model leaves on its outputs. It also employs an encoder and a decoder, both based on Convolutional Neural Networks (CNNs), to carry out the processes of watermark embedding and extraction.

## C.5 Standard DDPM and Improved DDPM.

**Standard DDPM.** Denoising Diffusion Probabilistic Model (DDPM), firstly developed by Ho et al. (2020), consists of a diffusion process $q\left(x_t \mid x_{t-1}\right)$ and a denoising process $p_\theta\left(x_{t-1} \mid x_t\right)$ which are respectively described as:

$$q\left(x_t \mid x_{t-1}\right) = \mathcal{N}\left(x_t; \sqrt{1-\beta_t}x_{t-1}, \beta_t I\right) \tag{11}$$

$$p_\theta\left(x_{t-1} \mid x_t\right) = \mathcal{N}\left(x_{t-1}; \mu_\theta\left(x_t, t\right), \Sigma_\theta\left(x_t, t\right)\right) \tag{12}$$

With the variance schedule $\beta_t$, a data point $x_0$ sampled from a real data distribution is transformed into noise $x_T$ by continuously adding a small amount of Gaussian noise to the sample for T steps. Then the image is gradually reconstructed by removing the noise from $x_T$ following the reverse diffusion process 12.

The most effective way to parameterize $\mu_\theta\left(x_t, t\right)$ is to predict the noise added to $x_0$ in each step with a neural network. In practice, we use the simplified objective suggested by Ho et al. (2020)

$$L_t^{\text{simple}} = \mathbb{E}_{t \sim [1,T], \mathbf{x}_0, \epsilon_t}\left[\left\|\boldsymbol{\epsilon}_t - \epsilon_\theta\left(\sqrt{\bar{\alpha}_t}\mathbf{x}_0 + \sqrt{1-\bar{\alpha}_t}\epsilon_t, t\right)\right\|^2\right]$$

Then the denoising process can be described as:

$$\mathbf{x}_{t-1} = \frac{1}{\sqrt{\alpha_t}} \left( \mathbf{x}_t - \frac{1-\alpha_t}{\sqrt{1-\bar{\alpha}_t}} \epsilon_\theta \left( \mathbf{x}_t, t \right) \right) + \sigma_t \mathbf{z}$$

**Improved DDPM.** Nichol & Dhariwal (2021) proposed a few modifications of DDPM to achieve faster sampling speed and better log-likelihoods. The primary modification is to turn $\Sigma_\theta \left( x_t, t \right)$ into a learned function using the formula

$$\Sigma_\theta \left( x_t, t \right) = \exp \left( v \log \beta_t + (1-v) \log \tilde{\beta}_t \right).$$

Moreover, they proposed a hybrid training objective

$$L_{\text{hybrid}} = L_t^{\text{simple}} + \lambda L_{\text{vlb}}$$

where $L_{\text{vlb}}$ refers to the variational lower-bound of DDPM. To reduce the variance of the training log loss of $L_{\text{vlb}}$, they proposed importance sampling:

$$L_{\text{vlb}} = E_{t \sim p_t} \left[ \frac{L_t}{p_t} \right], \quad \text{where } p_t \propto \sqrt{E \left[ L_t^2 \right]} \text{ and } \sum p_t = 1$$

Finally, they introduced an enhancement to the noise schedule with:

$$\bar{\alpha}_t = \frac{f(t)}{f(0)}, \quad f(t) = \cos \left( \frac{t/T + s}{1+s} \cdot \frac{\pi}{2} \right)^2$$

## D   ALGORITHM

As shown in Algorithm 2, the joint optimization is numerically solved by alternately training on the two levels. Every batch is first watermarked and trained on the classifier for upper level objective by gradient descent (line 4 to 6), and then optimized on basic patches for lower level objective by 5-step PGD (line 7 to 9). With the joint optimized basic patches and classifier, we can obtain a robust watermark that can encode different ownership information with a small change on the protected data. This watermark can be easily captured by the diffusion model and is effective for tracking data usage and copyright protection. The clean images $\{X_{1:n}\}$ for input of the algorithm is not necessary to be the images that we want to protect. The random cropped image blocks can help the basic patches to fit different image blocks and then increase the flexibility.

## E   EXAMPLES OF WATERMARKED IMAGES

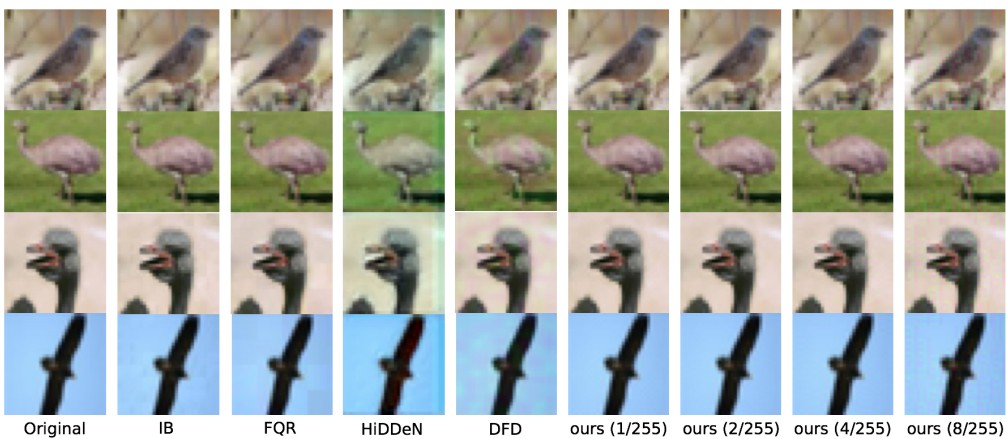

Figure 7: Examples of watermarked images of the bird class in CIFAR-10

---

**Algorithm 2** Joint optimization on $\{\boldsymbol{w}^{(1:B)}\}$ and $\mathcal{D}_\theta$

---

**Input:** Initialized basic patches $\{\boldsymbol{w}_{(0)}^{(1:B)}\}$, clean images $\{\boldsymbol{X}_{1:n}\}$, upper and lower level objectives in Eq. 4, $\mathcal{L}_{\text{upper}}$, $\mathcal{L}_{\text{lower}}$, watermark budget $\epsilon$, decoder learning rate $r$, batch size $bs$, PGD step $\alpha$ and epoch $E$.
**Output:** Optimal $\{\boldsymbol{w}^{(1:B),*}\}$ and $\theta^*$.

1: $step \leftarrow 0$
2: **for** $epoch$=1 to E **do**
3:     **for** $Batch$ from $\{\boldsymbol{X}_{1:n}\}$ **do**
4:         $\{\boldsymbol{p}_{1:bs}\} \leftarrow RandomCropBlock(Batch)$
5:         $\{\boldsymbol{w}_{1:bs}\}, \{\boldsymbol{b}_{1:bs}\} \leftarrow RandomPermutation(\{\boldsymbol{w}_{(step)}^{(1:B)}\}, bs)$
6:         $\theta \leftarrow StochasticGradientDescent(\frac{\partial \sum_1^{bs} \mathcal{L}_{\text{lower}}(\boldsymbol{p}_i + \boldsymbol{w}_i, \boldsymbol{b}_i, \theta)}{\partial \theta}, r)$     // Training on classifier
7:         **for** 1 to 5 **do**
8:             $\boldsymbol{w}_{(step)}^{(2:B)} \leftarrow Clip_{(-\epsilon, \epsilon)}\left(\boldsymbol{w}_{(step)}^{(2:B)} - \alpha sign(\frac{\partial \sum_1^{bs} \mathcal{L}_{\text{lower}}(\boldsymbol{p}_i + \boldsymbol{w}_i, \boldsymbol{b}_i, \theta)}{\partial \boldsymbol{w}_{(step)}^{(2:B)}})\right)$ // 5-step Projected Gradient Descent
9:         **end for**
10:        $step \leftarrow step + 1$
11:     **end for**
12:     return $\{\boldsymbol{w}_{(step)}^{(1:B)}\}$ and $\theta$.
13: **end for**

---

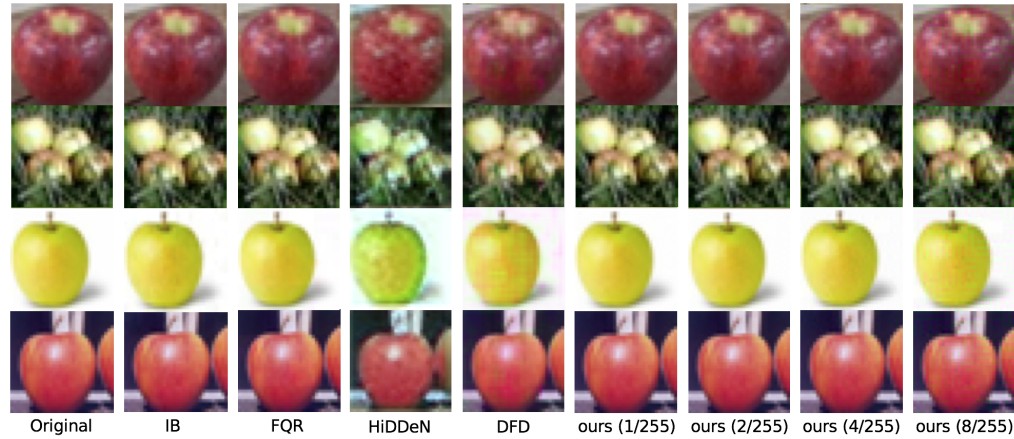

Figure 8: Examples of watermarked images of the apple class in CIFAR-100

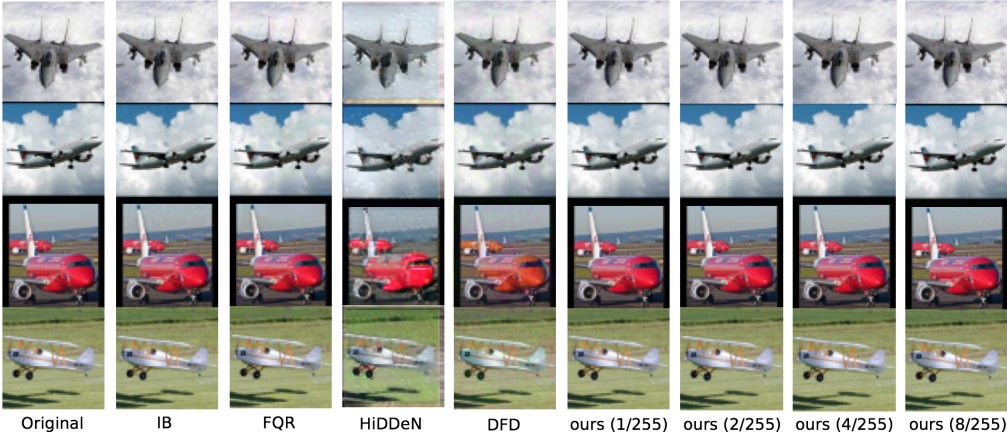

Figure 9: Examples of watermarked images of the plane class in STL-10

# F ADDITIONAL ANALYSIS ON THE INFLUENCE OF BUDGET AND WATERMARK RATE

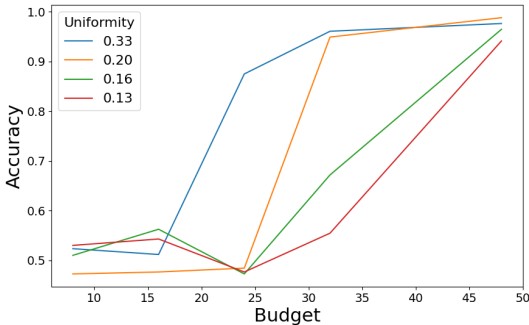

Figure 10: The change of bit accuracy under different budgets

As mentioned in Section 4.2, the reproduction of watermarks in generated images is related to the watermark budget and the watermark rate. In this subsection, we show that a larger budget and larger watermark rate can help with the reproduction of watermarks in the GDM-generated images.

In Figure 10, we follow the experimental setting in Section 3.2. We can see that when uniformity is the same, as the budget increases, the detection rate is also increasing, which means that watermarks can reproduce better if it has a larger budget. This can also be observed from Table 4.2 that the bit accuracy of budget 1/255 and 2/255 on CIFAR100 is lower than 4/255 and 8/255. Meanwhile, higher pattern uniformity can increase faster than lower pattern uniformity, which is consistent with Section 3.2.

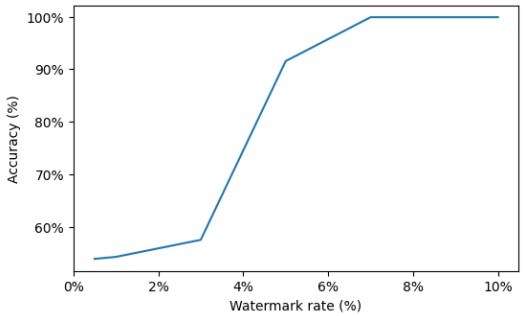

Figure 11: The change of bit accuracy with different watermark rates (budget=1/255)

In Figure 11, we follow the experimental setting in Section 4.1, while controlling the proportion of the watermarked images in the training set of GDM. From the figure, we can see that the bit accuracy on the generated images rises from about 53% to almost 100% when the watermark rate increases from 0.05% to 10%, which indicates that the watermark rate can affect the degree of reproduction of the watermark in generated images.

Figure 11 suggests that DiffusionShield cannot provide satisfied protection in the single-owner case when the watermark rate and the budget are small. In reality, the watermark rate for a single user may be small. However, there are multiple users who may adopt DiffusionShield to protect the copyright of their data. Therefore, next we check how the performance of DiffusionShield changes with the number of users when the watermark rate and the budget are small for each user. Although each user has a distinct set of watermarked data, they all share the same set of basic patches, which has the potential to enhance the reproducibility of the watermark. As shown in Figure 12, we have $K$ owners and the images of each owner compose 1% of the collected training data. As the number of owners increases from 1 to 20, the average accuracy increases from about 64% to nearly 100%. This observation indicates that DiffusionShield can work with multiple users even when the watermark rate

Table 5: Bit accuracy (%) with speeding-up models

| $l_\infty$ | | CIFAR10 | CIFAR100 | STL10 |
|---|---|---|---|---|
| 1/255 | Cond. | 99.7824 | 52.4813 | 95.8041 |
| | Uncond. | 94.6761 | 52.2693 | 82.4564 |
| 2/255 | Cond. | 99.9914 | 64.5070 | 99.8299 |
| | Uncond. | 96.1927 | 53.4493 | 90.4317 |
| 4/255 | Cond. | 99.9996 | 99.8445 | 99.9102 |
| | Uncond. | 96.1314 | 92.3109 | 95.7027 |
| 8/255 | Cond. | 100.0000 | 99.9984 | 99.9885 |
| | Uncond. | 95.7021 | 92.2341 | 95.3009 |

and the budget are small for each user. Since GDM often collects training data from multiple users, this study suggests that DiffusionShield could be very practical.

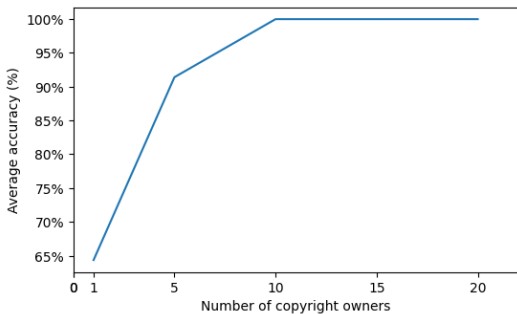

Figure 12: The change of bit accuracy with different numbers of copyright owners (budget=2/255)

# G    ADDITIONAL EXPERIMENTS ON ROBUSTNESS

## G.1    ROBUSTNESS UNDER SPEEDING-UP SAMPLING MODELS

Speeding-up sampling is often employed by practical GDMs due to the time-consuming nature of the complete sampling process, which requires thousands of steps. However, the quality of the images generated via speeded-up methods, such as Denoising Diffusion Implicit Model (DDIM) (Song et al., 2020), is typically lower than normal sampling, which could destroy the watermarks on the generated images. In Table 5, we show the performance of DiffusionShield with DDIM to demonstrate its robustness against speeding-up sampling. Although DiffusionShield has low accuracy on CIFAR100 when the budget is 1/255 and 2/255 (same as the situation in Section 4.2), it can maintain high accuracy on all the other budgets and datasets. Even with a 1/255 $l_\infty$ budget, the accuracy of DiffusionShield on CIFAR10 is still more than 99.7% in class-conditionally generated images and more than 94.6% in unconditionally generated images. This is because the easy-to-learn uniform patterns are learned by GDMs prior to other diverse semantic features like shape and textures. Thus, as long as DDIM can generate images with normal semantic features, our watermark can be reproduced in these images.

## G.2    ROBUSTNESS UNDER DIFFERENT HYPER-PARAMETERS IN TRAINING GDMS

Besides the speeded up sampling method, we test two more hyperparameters in Table 6 below. They are learning rate and diffusion noise schedule. Diffusion noise schedule is a hyperparameter that controls how the gaussian noise added into the image increases during the diffusion process. We test with two different schedules, cosine and linear. We use DiffusionShield with 2/255 budget to protect one class in CIFAR10. The results show that the watermark accuracies in all the different parameters are higher than 99.99%, which means our method is robust under different diffusion model hyperparameters.

Table 6: Bit accuracy under different hyper-parameters of DDPM

|       | cosine    | linear    |
| ----- | --------- | --------- |
| 5e-4  | 99.9985%  | 99.9954%  |
| 1e-4  | 99.9945%  | 99.9908%  |
| 1e-5  | 99.9939%  | 99.9390%  |

## H   DETAILS OF GENERALIZATION TO FINE-TUNING GDMs

In Table 7 and Table 8, we measure the generated quality of both watermarked class and all classes to show that DiffusionShield will not influence the quality of generated images. We use FID to measure the quality of generated images. Lower FID means better generated quality. Comparing FIDs of watermarked classes by different watermark methods, we can find that our method can keep a smaller FID than DFD and HiDDeN when the budget is smaller than 4/255. This means our watermark is more invisible. Comparing FID of ours and clean data, we can find that our method has almost no influence on the generated quality of GDMs. We can also see that FID for the watermarked class is usually higher than FID for all the classes. This is because FID is usually lager when the sample size is small and we sample fewer images in watermarked class than the total number of the samples from all the classes. In summary, our method will not influence the quality of generated images.

Table 7: Bit accuracy under different hyper-parameters of DDPM

| method | clean  | ours(1/255) | ours(4/255) | ours(8/255) | DFD    | HiDDeN |
| ------ | ------ | ----------- | ----------- | ----------- | ------ | ------ |
| FID    | 15.633 | 14.424      | 26.868      | 51.027      | 33.884 | 48.939 |

Table 8: Bit accuracy under different hyper-parameters of DDPM

| method | clean | ours(1/255) | ours(4/255) | ours(8/255) |
| ------ | ----- | ----------- | ----------- | ----------- |
| FID    | 3.178 | 4.254       | 3.926       | 4.082       |

## I   DETAILS OF GENERALIZATION TO FINE-TUNING GDMs

**Background in fine-tuning GDMs.** To speed up the generation of high-resolution image, Latent Diffusion Model proposes to project the images to a vector in the hidden space by a pre-trained autoencoder (Rombach et al., 2022). It uses the diffusion model to learn the data distribution in hidden space, and generate images by sampling a hidden vector and project it back to the image space. This model requires large dataset for pre-training and is commonly used for fine-tuning scenarios because of the good performance in pre-trained model and fast training speed of fine-tuning.

**Generalization to fine-tuning GDMs.** To use our method and enhance the pattern uniformity in the fine-tuning settings, we make two modifications. 1) In stead of enhancing the uniformity in pixel space, we add and optimize the watermark in hidden space and enhance the uniformity in hidden space. 2) Instead of using PGD to limit the budget, we add $l_2$ norm as a penalty in our objective.

**Experiment details.** We use the *pokemon-blip-captions* dataset as the protected images and following the default settings in *huggingface/diffusers/examples/text_to_image* (von Platen et al., 2022) to finetune a Stable Diffusion, which is one of Latent Diffusion Models.

## J    VISUALIZATION

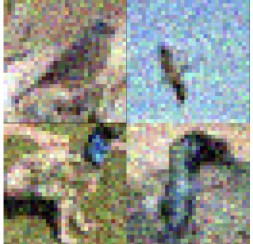

(Mean =0 Variance=0.1)

Figure 13: The Gaussian noise added to the images in the experiments in Table 4.

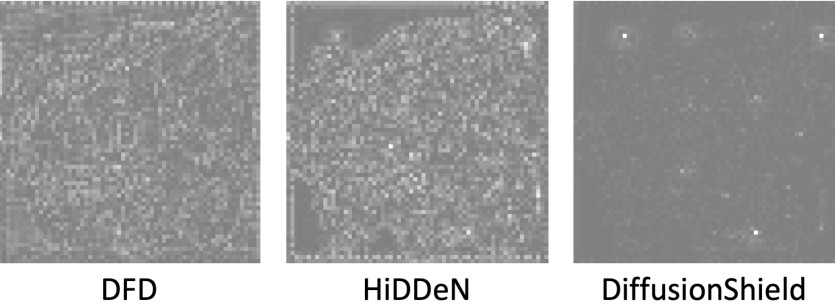

Figure 14: The change of hidden space after watermarking.

**Visualization of the Gaussian noise added to the images in the experiments in Table 4.** In Figure 13, we visualize the change of hidden space. The hidden space of SD is in shape of [4, 64, 64] which has 4 channels. We visualize the Gaussian noise which is added to the images in the experiments in Table 4. The variance of the Gaussian Noise is 0.1.

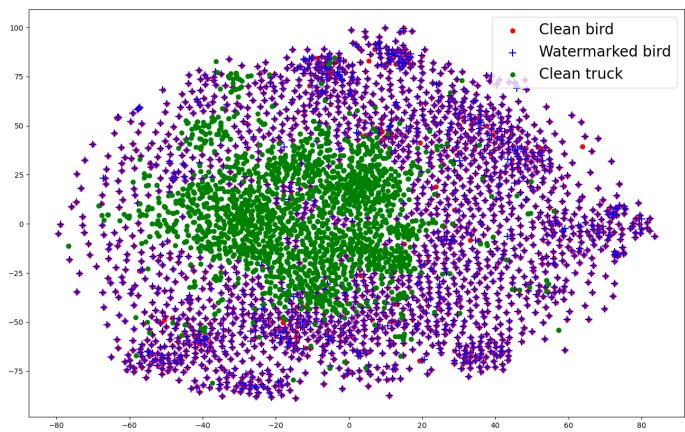

Figure 15: The change of hidden space after watermarking.

**Visualization of hidden space of Stable Diffusion.** In Figure 14, we visualize the change of hidden space. The hidden space of SD is in shape of [4, 64, 64] which has 4 channels. We visualize one of channel and find that the change of DFD and HiDDeN is much obvious than ours.

**Visualization of feature space extracted by Contrastive Learning** In Figure 15, we visualize the influence of watermark on the feature space. We use Contrastive Learning Chen et al. (2020) to extract the feature of both clean and watermarked class.

