# OpenReview forum: "DiffusionShield: A Watermark for Data Copyright Protection against Generative Diffusion Models"
_ICLR.cc/2024/Conference — Submitted to ICLR 2024_

### Official Review · Reviewer_2t92 · 2023-10-23

**Soundness:** 3 good
**Presentation:** 3 good
**Contribution:** 3 good
**Rating:** 5
**Confidence:** 3

**Summary:**

To address the potential copyright infringement caused by Generative Diffusion Models (GDMs), especially the unauthorized imitation of styles and appearances of the products from artists and fashion companies, the authors propose a novel watermarking technique called DiffusionShield. This model embeds secret messages into protected images in an imperceptible manner, ensuring that these watermarks are easily learned by GDMs and will be reproduced in other generated images. Existing watermarking techniques were not specifically designed for GDMs, making them either hard for GDMs to learn or requiring larger perturbations to maintain partial watermarks. In contrast, the authors develop a new watermarking technique tailored for GDMs, based on the finding that GDMs more easily learn watermarks with high pattern uniformity and theoretical prove that uniform patterns are prioritized in learning over the original images. To capitalize on this pattern uniformity, DiffusionShield introduces a blockwise strategy and a joint optimization method that not only further enhance detection accuracy but also reduce the budget. Finally, the authors validate their model’s superiority over existing methods in terms of visual quality and detection accuracy through various experiments, and demonstrating its effectiveness in both single-user and multi-user scenarios.

**Strengths:**

1. The article organizes the problem statement in a clear manner, offering a comprehensive discussion on protection scenarios, including both single-owner and multiple-owner cases.
2. The authors properly define "pattern uniformity"(Equation 1) and provide related theoretical explanation. They further conduct experiments to substantiate the significance of this concept.
3. The experiments are extensive, examining the influence of different Perturbation Budgets, Message Lengths, and Watermark Rates on bit accuracy. Additionally, in a multiple-owner scenario, the paper discusses the impact of multiple users on bit accuracy.
4. The performance is good. The proposed method achieves state-of-the-art bit accuracy and perturbation budget across multiple datasets and maintains superior robustness under various image distortions (except for gaussian noise).

**Weaknesses:**

1. The paper carries out extensive experiments but could improve in terms of layout. A considerable amount of experimental data is placed in the appendix, with minimal mention in the main text. As a potential user of this model, I would be particularly concerned about the issue of watermark rates. Specifically, it's worth questioning whether the proposed method remains effective when the training dataset contains only a limited number of watermarked images. Appendix F's Figure 11 briefly touches upon this issue, but this aspect should be more explicitly mentioned in the main text.

2. The paper employs the HiDDeN model for deep learning-based watermarking, which is no longer state-of-the-art (SOTA). Nevertheless, HiDDeN outperforms the proposed method in certain experiments, such as under the 'Uncond.' condition in Table 1 for CIFAR10 and in the  distortion method of Gaussian noise in Table 4. Therefore, the authors should utilize a state-of-the-art deep learning-based watermarking model to further substantiate the superiority of their method.

3. In the Implementation Details section, the authors build training dataset by "designating one random class of images as watermarked images, while treating other classes as unprotected images." While using images from the same class might be an ideal and simpler scenario that could further promote pattern uniformity, in real-world applications, images could come from various classes. The authors need to elaborate more on the effectiveness of their method under such circumstances.

**Questions:**

1. For the first weakness, I understand that due to the constraints of the main text's length, it might be challenging for the authors to include all experimental details. However, I would still suggest at least mentioning the experiment of watermark rates in the main body of the paper. This is merely a suggestion on my part; even if the authors choose not to adopt it, it won't lead me to lower my rating of the paper.

2. The second point of weakness is a significant concern for me. I hope the authors can provide direct experiments to alleviate my doubts. The state-of-the-art (SOTA) model that I'm currently aware of is "Towards Blind Watermarking: Combining Invertible and Non-invertible Mechanisms."

3. For the third weakness, I believe the authors need to conduct more comprehensive experiments to further substantiate the effectiveness of their method across multiple real-world scenarios.

---

> ### Author Response · Authors · 2023-11-16
> **Response to reviewer 2t92**
>
> Thank you very much for the valuable feedback. We hope the following responses can help address your concerns.
>
> > **Weakness 1 & Question 1**: Improvement on the layout by incorporating the analysis on the watermark rate.
>
> **Response**: Thanks for your suggestion about the paper organization. We updated the main text of Section 4.2 to incorporate the discussion on the influence of watermark rate. In the main text, we point out that, although the low watermark rate might hurt the performance, this problem is mitigated in the multi-user case. Even though the watermark rate for each user is low and they encode different messages and do not share the pattern uniformity, our method can still perform well. Then we refer the readers to read more details about the experiment in Appendix F. The updated part is marked blue in the revision for your convenience.
>
> > **Weakness 2 & Question 2**: More state-of-the-art deep learning-based watermarking models should be considered as the baselines for comparison.
>
> **Response**: Thanks for your suggestion regarding the inclusion of new baselines. In response, we have expanded our comparative analysis to include three additional baselines: IGA[1], MBRS[2], and CIN[3]. The results are reported in Table 15 below. We can see that the performance of IGA is very similar to HiDDeN and DFD. Although IGA’s bit accuracy is comparable to our DiffusionShield, it demands a significantly higher budget—nearly 10 times the Linf and LPIPS values of our approach. As for MBRS and CIN, despite having budgets similar to DiffusionShield, their bit accuracies are worse than ours, especially on CIFAR100, where MBRS achieves only 87.68% and CIN is 51.13% in bit accuracy. In contrast, DiffusionShield maintains the bit accuracy of 99.80% with a comparable budget. This is because of the higher pattern uniformity of DiffusionShield. In summary, the performances of the new baselines are similar to the previous baseline methods. They either compromise the budget for bit accuracy close to ours, or fail to be reproduced well in the generated images.
>
> [*Table 15*]
> |Dataset|budget|IGA|MBRS|CIN|ours|
> |-|-|-|-|-|-|
> |CIFAR10|$L_{\infty}$|52/255|16/255|8/255|4/255|
> ||$L_{2}$|3.38|0.36|0.42|0.72|
> ||LPIPS|0.08910|0.00182|0.00185|0.00720|
> ||Bit Acc.(Cond.)|99.63%|99.97%|99.97%|100.00%|
> ||Uniformity|0.063|0.518|0.599|0.964|
> |CIFAR100|$L_{\infty}$|66/255|19/255|9/255|4/255|
> ||$L_{2}$|5.31|0.43|0.44|0.72|
> ||LPIPS|0.08830|0.00129|0.00105|0.00134|
> ||Bit Acc.(Cond.)|97.25%|87.68%|51.13%|99.80%|
> ||Uniformity|0.162|0.394|0.527|0.915|
>
> [1] Zhang, Honglei, et al. Robust data hiding using inverse gradient attention.
>
> [2] Jia et al. Mbrs: Enhancing robustness of dnn-based watermarking by mini-batch of real and simulated jpeg compression, ACM MM '21
>
> [3] Ma et al. Towards Blind Watermarking: Combining Invertible and Non-invertible Mechanisms, ACM MM '22
>
> > **Weakness 3 & Question 3**: More comprehensive experiments to further substantiate the effectiveness of the proposed method across multiple real-world scenarios, especially when the protected images are from various classes instead of one class.
> **Response**: Thanks for your suggestions on the implementation details. In response to your concerns, we have added one set of experiments using CIFAR-100. In the following Table 16, we have considered that the protected images are from various classes (5, 10 or 15 classes randomly selected in CIFAR-100) and added watermarks to those protected classes. The bit accuracy of the watermark on the generated images from the protected class are shown in the table. It reveals that the watermarks are predominantly detectable in the images across various classes, with limited exceptions noted in scenarios where the watermark budget is minimal (1/255) and only 5 classes are watermarked.
>
> [Table 16]
> |Budget|5 class|10 class|15 class|
> |-|-|-|-|
> |8/255|100.000%|99.995%|99.994%|
> |4/255|99.998%|99.980%|99.993%|
> |2/255|99.748%|99.973%|99.969%|
> |1/255|65.525%|98.824%|99.707%|

---

> > ### Comment · Reviewer_2t92 · 2023-11-21
> >
> > The authors have satisfactorily addressed my concerns regarding the first and third questions. For the third question, extensive experiments demonstrate the effectiveness of the authors' method across various practical scenarios.
> >
> > However, regarding the second question, the authors' experiments have increased my concerns. Firstly, in the CIFAR-10 dataset, the authors' method only leads in L_infinite in terms of image quality, without outperforming MBRS and CIN in L_2 and LPIPS. Additionally, the lead in accuracy (ACC) is limited. The superiority in Uniformity doesn't seem to bring the “positive correlation between the watermark detection rate in the GDM-generated images and the pattern uniformity” as claimed in the paper, potentially challenging the foundation of the article.
> >
> > In CIFAR-100, the performance of CIN is peculiar, with an accuracy of 51.13% resembling random guessing in the context of binary watermarks. This needs clarification from the authors. Also, CIN and MBRS were originally tested on datasets with a higher resolution, and the original paper includes tests on the STL10 and ImageNet-20 datasets, but these are missing in the rebuttal.
> >
> > Based on these points, I will maintain my score.

---

> ### Author Response · Authors · 2023-11-17
> **A friendly reminder**
>
> We are grateful for the useful comments provided by you. We hope that our answers have addressed your concerns. If you have any further concerns, please let us know. We are looking forward to hearing from you.

---

> ### Author Response · Authors · 2023-11-22
> **Response to reviewer 2t92 (1/2)**
>
> Thanks for providing your insightful feedback.  We hope that our following answers can address your concerns.
>
> **Regarding the performance on CIFAR10:**
> - *1) Our method does not outperform MBRS and CIN in terms of image quality.*
>
>     Thanks for pointing out the concerns. We note that there is a typo about LPIPS budget of our method for CIFAR-10 in the last response. It should be **0.00120** instead of 0.00720, which is actually smaller than the baselines. We have fixed the typo in the following table. In addition, considering that our method provides a flexibility to control the budget of the watermark, we further demonstrate the performance of our methods considering lower L-inf budget, 2/255 and 1/255 in the table below. The new results, presented in the table below, indicate that our method maintains comparable bit accuracy with the baseline methods even at much reduced budgets. Notably, with an L-inf budget of 1/255, our method achieves a bit accuracy of 99.90%, while both the L_2 and LPIPS metrics are significantly lower than those of the baseline methods. These findings underscore the effectiveness of our approach when working with smaller budgets.
>
> - *2) Uniformity doesn't seem to bring the positive correlation between the watermark detection rate and the pattern uniformity.*
>
>     The effectiveness of the pattern uniformity is more outstanding when the budget is lower and the scenario is more demanding (e.g. when watermark rate is low). The scenario considered in the experiments on CIFAR10 is relatively easier given that the watermark rate (10%) is relatively high, making the watermarks easier to be learned by the GDM and reproduced in the generated images. That is the reason why the baseline methods can also achieve good performance on the CIFAR10 dataset. But we can see the superiority of pattern uniformity in reaching a lower budget in CIFAR10 from the point 1 above. Also, in the following discussion regarding CIFAR100 we will show that if the scenario is more demanding, the importance of the pattern uniformity will be more obvious.
>
> |Dataset|budget|IGA|MBRS|CIN|ours|ours|ours|
> |-|-|-|-|-|-|-|-|
> | CIFAR10|Linf|52/255|16/255|8/255|**1/255**|2/255|4/255|
> ||L2|3.38|0.36|0.42|**0.18**|0.36|0.72|
> ||LPIPS|0.08910|0.00182|0.00185|**0.00005**|0.00020|0.00120|
> ||Cond.|99.63%|99.97%|99.97%|99.90%|99.99%|**100.00%**|
> ||Uniformity|0.063|0.518|0.599|0.974|0.971|0.964|
>
> **Regarding the performance on CIFAR100 dataset:**
> - *1) Why is the performance of CIN peculiar in CIFAR100?*
>
>     As we discussed, the reproduction of the watermark is related to factors including pattern uniformity, budget and watermark rate. In our experiments, we consider adding watermarks to the images from one certain class. Thus, the watermark rate for CIFAR100 (1%) is much lower compared with CIFAR10 (10%). Therefore, CIFAR-100 is a more difficult dataset. Both FRQ and CIN fail to protect in this dataset and have only an accuracy similar to random guess result. They do not have very extremely large budget like HiDDeN and do not have good pattern uniformity than DiffusionShield. That is the reason why CIN suffers from an obvious degradation from CIFAR10 to CIFAR100 taking the three factors (watermark rate, budget  and pattern uniformity) into consideration.
>
> - *2) The superiority of our method in CIFAR100 stems from the high pattern uniformity.*
>
>     Given the inherent complexity of CIFAR100 and the fact that the watermarking rate is limited to just 1%, it becomes significantly challenging for the watermark to be replicated accurately in the generated images. In more demanding scenarios, the advantage of our highly uniform watermark pattern becomes more evident. That is the reason why our method demonstrates a very good performance in CIFAR100. This further highlights the necessity of high pattern uniformity, particularly in practical applications where complex data sets and low watermark rate are involved.

---

> ### Author Response · Authors · 2023-11-22
> **Response to reviewer 2t92 (2/2)**
>
> **Regrading the results on STL10 and ImageNet-20 datasets:**
>
> In response to your concerns, we have also conducted the experiments on STL10 and ImageNet-20 datasets. Given that training a diffusion model on images with higher resolutions requires a very high computational cost, it takes much longer time to get the results. We will update the results in the latest version of the paper once we get them.
>
> Based on the previous results on images with size from 32x32 to 256x256, we can deduce certain trends regarding performance changes of watermarking methods as image resolution increases. We summarize the trend of the change in the bit accuracy of our method on different datasets below. From the table below we can see that when the size of the images is increased across different datasets, there is no significant change in the bit accuracy in our method (4/255). But the baselines (including CIN and MBRS) reduce in a demanding scenario. Therefore, although we haven’t got the results for the experiments with higher resolutions of STL10 and ImageNet20, we believe that following this trend, when the image resolution continues to increase like STL10 and ImageNet (which is usually a more demanding scenario), our DiffusionShield is very likely to keep a higher performance than MBRS and CIN.
>
> |Dataset|CIFAR10|CIFAR100|STL10|ImageNet|
> |-|-|-|-|-|
> |ours (4/255)|100.00%|99.80%|99.89%|99.95%|
> |MBRS| 99.97%|87.68%|||
> |CIN| 99.97%|51.13%|||

---

> > ### Author Response · Authors · 2023-11-22
> > **Looking forward to your reply**
> >
> > Thanks again for your feedback. We hope that our answers have addressed your concerns. If you have any further concerns, please feel free to let us know. We are pleasant to address all your concerns during the discussion period.

---

> > > ### Author Response · Authors · 2023-11-23
> > > **A friendly reminder**
> > >
> > > We appreciate your reviews. We hope that our responses have adequately addressed your concerns. Should you have any additional questions or points you'd like to discuss, please feel free to let us know. We are keen to engage in further discussion.

---

### Official Review · Reviewer_NoWK · 2023-11-01

**Soundness:** 4 excellent
**Presentation:** 4 excellent
**Contribution:** 4 excellent
**Rating:** 8
**Confidence:** 5

**Summary:**

This paper tackles the problem of Generative Diffusion Models (GDM) using unauthorized images as training data and learns the style/distribution of those unauthorized images.

This paper introduces DiffusionShield, a watermark to protect data copyright. It is motivated by the observation that "pattern uniformity" can effectively assist the watermark to be captured by GDMs. This pattern uniformity is how uniform the watermark patterns are. That is, lower variance in watermark patterns can cause GDMs to learn watermark features before the actual features.

By enhancing the pattern uniformity of watermarks and leveraging a joint optimization method, DiffusionShield successfully secures copyright with better accuracy and a smaller budget. Theoretic analysis and experimental results demonstrate the superior performance of DiffusionShield

**Strengths:**

originality: the paper proposes a novel approach to watermarking the images to prevent malicious GDMs from using the training data without authorization. The method is simple and effective, supported by theoretical and experimental results.

quality: this paper is technically sound, both theoretical and experimental results are provided to support the claims.

clarity: this paper is well-written and well-organized. the presentation is great.

significance: this paper is significant and addresses a very important problem in the field of GDMs -- data copyright protecting. GDMs nowadays are strong enough to learn any data distribution and able to replicate some novel images using the style of the creator without crediting. This is because there is no proof that the GDMs have used the images from the creator. This paper proposes a simple but robust method to help protect the creator's copyright on the data they own.

**Weaknesses:**

I notice this method mostly describes adding a watermark to the original images instead of the latent space which is commonly used by existing GDMs such as Stable Diffusion (SD). Although there is a section describing fine-tuning a SD and adding watermarks to the latent space, there is much less visualization/analysis on it. The author can provide more examples/analysis of adding a watermark to latent space and how it will affect the VAE encoding/decoding process.

Further, the author can provide the cost of watermarking the images and it maybe not be likely that a content creator has the resources to watermark the images.

Finally, I think there is less analysis on how DiffusionShield can protect against the removal attack --- if the malicious took the watermarked data, perform some augmentations on the data. Will the watermark still able to be detected?

**Questions:**

1. If the malicious took the watermarked data, perform some augmentations on the data. Will the watermark still able to be detected?
2. How about the feature distribution of the watermarked images? Although little pixel changes are not visible to the human eyes but may be more obvious in the feature distribution of some feature extractors.

---

> ### Author Response · Authors · 2023-11-16
> **Response to reviewer NoWK (1/2)**
>
> Thanks for your positive comments. Please find our response below for your questions.
>
> > **Weakness 1**: The author can provide more examples/analysis of adding a watermark to latent space and how it will affect the VAE encoding/decoding process.
>
> **Response**: Thanks for your insightful query regarding the influence of watermark in latent space. To address this, we compared the budget and visualized the change of latent space in updated Appendix J. As shown in the below Table 13, in latent space, the budget of baselines including both DFD and HiDDeN are much larger than our method. Even though the baseline methods have such large budgets, we know that, from Table 3 in Section 4.4, their performance of protection is still worse than ours. (For your convenience, we also repeat Table 3 below. ) Also, we visualize the change of latent space after watermarking in Appendix J and find that the change of DFD and HiDDeN is much more obvious than ours. Regarding the VAE, after we decode from the watermarked latent space, the budget is still small and the watermark is imperceptible in pixel space. This means that our watermark has a much smaller influence on the decoding process compared with baseline methods.
>
> [*Table 13*]
> ||DFD|HiDDeN|ours|
> |-|-|-|-|
> |latent space|191.3678|202.1912|42.1265|
> |pixel space|88.9103|81.2732|31.2287|
>
> > **Weakness 2**: The author can provide the cost of watermarking the images and it may not be likely that a content creator has the resources to watermark the images.
> >
> **Response**: Thanks for your helpful question. The training process of watermark patches and classifiers requires about 10 hours on a single V100 GPU card. For content creators, even if they have no access to GPU resources for training, they can reuse the off-the-shelf basic patches and classifier to protect their artworks. This is one of our  advantages, flexibility.  As we mentioned in Section 3.1, 3.3 and Experiment in Section 4.3, once the watermarks are obtained, DiffusionShield does not require retraining when there is an influx of new users and images. Therefore, a third-party agency can train these patches and classifiers and provide the service. The content creators can entrust the third-party agency to watermark and validate it.
>
> > **Weakness 3 & Question 1**: Analysis on how DiffusionShield can protect against the removal attack
>
> **Response**: Thanks for your question. In Table 4 in Section 4.6, we have conducted experiments to demonstrate the robustness of our method against various forms of image corruption including Gaussian noise, low-pass filtering, grayscale, and JPEG compression. In order to further clarify your concern about the robustness, we have expanded the scope of our experiments to include image resizing and a deep-learning-based watermark removal method [1]. These tests are conducted using the CIFAR-10 dataset. For the resizing experiments, we altered image sizes from 32x32 to 64x64 (termed "large" in the table), or from 32x32 to 16x16 (termed "small" in the table). During detection, we resize all the data back to 32x32 before inputting them to the detector. In order to improve the robustness, we have incorporated the corresponding disturbances as an augmentation in the training of the watermark. All the results related to robustness are put together in the following Table 14. From the results we can see that DiffusionShield consistently achieves a high bit accuracy and outperforms the two baseline methods in all scenarios except under the Gaussian Noise corruption. In summary, the results from these experiments indicate that our DiffuisonShield method is able to maintain robustness against multiple disturbances.
>
> [*Table 14*]
> ||DFD|HiDDeN|Ours|
> |-|-|-|-|
> |No corruption|93.57%|98.93%|99.99%|
> |Resize (Small)|92.38%|83.13%|99.30%|
> |Resize (Large)|93.20%|79.69%|99.99%|
> |Watermark Removal [1]|91.11%|82.20%|99.95%|
> |Gaussian Noise|68.63%|83.59%|81.93%|
> |Low-pass filter|88.94%|81.05%|99.86%|
> |Greyscale|50.82%|97.81%|99.81%|
> |JPEG Comp.|62.52%|74.845|94.45%|
>
> [1] Invisible Image Watermarks Are Provably Removable Using Generative AI. Zhao et. al., 2023

---

> ### Author Response · Authors · 2023-11-16
> **Response to reviewer NoWK (2/2)**
>
> > **Question 2**: The feature distribution of the watermarked images.
>
> **Response**:  Thanks for your suggestion regarding the feature distribution of watermarked images. To address this, we employed a Resnet18 model pre-trained by Contrastive Learning as our feature extractor to check the change of feature distribution after watermarking. We visualize the feature space by t-SNE with the updated results now included in Figure 15 in Appendix J. In this visualization, we plot three clusters: the clean bird class (marked as red dots), the bird class watermarked by DiffusionShield with a budget of 8/255 (represented by blue “+” symbols), and the clean truck class (indicated by green dots).  From the figure we can see that, watermarked bird class exhibits negligible change compared to the clean bird class. Each sample in clean bird class shares the same coordinate with a watermarked bird sample. This indicates that the watermark has no influence on the feature space. For comparison, we also included the truck class in our visualization, which demonstrates a distinctly different distribution compared to the two bird classes. In summary, our watermark has almost no influence on not only the visibility but also the underlying feature distribution.

---

> ### Author Response · Authors · 2023-11-17
> **A friendly reminder**
>
> We are grateful for the useful comments provided by you. We hope that our answers have addressed your concerns. If you have any further concerns, please let us know. We are looking forward to hearing from you.

---

> ### Comment · Reviewer_NoWK · 2023-11-22
>
> I have carefully read the authors responses and they have addressed my concerns. I also appreciate the authors effort. I am staying at my rating due to the reason I stated in the strengths.

---

> > ### Author Response · Authors · 2023-11-22
> > **Response to reviewer NoWK**
> >
> > We are glad to hear that! Thank you very much for your feedback.

---

### Official Review · Reviewer_Bfvt · 2023-11-04

**Soundness:** 3 good
**Presentation:** 3 good
**Contribution:** 2 fair
**Rating:** 3
**Confidence:** 4

**Summary:**

This paper introduces DiffusionShield, a watermark to protect data copyright, which adopt the "pattern uniformity" to assist the watermark to be captured by GDMs. The current experiments demonstrate the effectiveness of the proposed method. However, I think the method is not practical enough.

**Strengths:**

- The idea of this paper is intersting. Addressing the root cause of infringement (i.e., watermarking released data) can make copyright protection more comprehensive and reliable.

**Weaknesses:**

- Details of parameter settings are best placed inside the main paper rather than in supplementary material. Besides, the details of the compared methods should give. For example, how to reproduce the HiDDeN---the noise layer follows the paper? If so, it may not be fair since there are no attack (e.g., JPEG compression) introduced in the experiment.
- The setting of the experments are not practical.
   - should defend against possible attack. At least, the author should consider JPEG compression and Resize attack which is very common in the real scenario. If the generated images are resized to other shape, does this method still work?
   - should introduced high resolution images ([1] is a good example). I don't think it makes sense to validate with low resolution images because GDM usually generates higher resolution images.

[1] Guo et al. practical deep dispersed watermarking with synchronization and fusion, ACM MM'23

**Questions:**

- It seems that each 4x4 basic block can represent 2bit (if B=4), so how to embed 128bit in CIFAR-10? For me, the message should be (28/4)x(28/4)x2bit=98bit, right?

---

> ### Author Response · Authors · 2023-11-16
> **Response to reviewer Bfvt**
>
> Thanks for your valuable comments. Please find our response below for your questions.
>
> > **Weakness 1**: Details of parameter settings are best placed inside the main paper. The missing details of the implementation of baseline HiDDeN (whether it has the noise layers) may cause misunderstanding about the fairness of experiment settings.
>
> **Response**: Thanks for your helpful suggestion. Following your suggestions, we have placed the details of parameters settings in Section 4.1 in the main paper. Regarding the implementation of HiDDeN, our settings are as fair as you expected. Specifically, in our experiments, we did not include the noise layers in HiDDeN except when testing for robustness against noise. This means for all experiments other than those reported in Table 4 of Section 4.6, HiDDeN was used without its noise layer. However, for experiments focused on testing robustness, we did incorporate a noise layer in HiDDeN to maintain fairness.  We have updated the details of this baseline setting in Appendix C4 and marked the update part as blue color in the revision for your convenience. We hope this update clears up any confusion regarding the baseline settings in our experiments.
>
> > **Weakness 2**: Experiments on the watermark's robustness against possible attack such as JPEG compression and resize are necessary.
>
> **Response**: Thanks for your suggestions about the robustness analysis. In Table 4 of Section 4.6, we have conducted experiments to demonstrate our robustness against attack including JPEG compression, Gaussian noise, low-pass filter and grayscale. To further clarify your concern about the robustness, we add more experiments about resize and a deep-learning-based watermark removal method [1]. In the following Table 12, we merge all the results including Table 4 for comparison. These tests are conducted using the CIFAR-10 dataset. For the resizing experiments, we altered image sizes from 32x32 to 64x64 (termed "large" in the table), or from 32x32 to 16x16 (termed "small" in the table). During detection, we resize all the data back to 32x32 before inputting them to the detector. In order to improve the robustness, we have incorporated the corresponding disturbances as an augmentation in the training of the watermark. Notably, DiffusionSheild maintains bit accuracy above 94% and outperforms the two baseline methods in all scenarios except under the Gaussian Noise corruption. In comparison, DFD suffers from a significant reduction in Gaussian noise, greyscale and JPEG compression, and HiDDeN shows a poor performance under low-pass filter, resize, watermark removal attack, and JPEG Compression. In summary, the results from these experiments indicate that our DiffuisonShield method is able to maintain robustness against multiple disturbances.
>
> [1] Invisible Image Watermarks Are Provably Removable Using Generative AI. Zhao et. al., 2023
>
> [*Table 12*]
> ||DFD|HiDDeN|Ours|
> |-|-|-|-|
> |No corruption|93.57%|98.93%|99.99%|
> |Resize (Small)|92.38%|83.13%|99.30%|
> |Resize (Large)|93.20%|79.69%|99.99%|
> |Watermark Removal [1]|91.11%|82.20%|99.95%|
> |Gaussian Noise|68.63%|83.59%|81.93%|
> |Low-pass filter|88.94%|81.05%|99.86%|
> |Greyscale|50.82%|97.81%|99.81%|
> |JPEG Comp.|62.52%|74.85%|94.45%|
>
> > **Weakness 3**: Experiments on high-resolution images are necessary.
>
> **Response**: Thanks for your suggestion. The experiments of high-resolution images are reasonable and necessary and we do have ImageNet in Table 1 in Section 4.2 and pokemon-blip-captions in Table 3 in Section 4.4. ImageNet is in shape 256x256, and pokemon-blip-captions is in 512x512. These datasets have similar resolution as the dataset in the suggested paper [1], and they are also similar to most of Diffusion Model papers, like [2] [3], whose experiments are mainly conducted with CIFAR-10 in 32x32, ImageNet in 64x64 and LSUN in 256x256, and Latent Diffusion papers like [4], whose experiments are mainly conducted using image with size 512x512 and 256x256. We hope this can clarify your concern about the image resolution.
>
> [1] Guo et al. Practical deep dispersed watermarking with synchronization and fusion, ACM MM'23
>
> [2] Ho et al. Denoising Diffusion Probabilistic Models, NeurIPS’ 20
>
> [3] Nichol et al. Improved Denoising Diffusion Probabilistic Models, ICML’ 21
>
> [4] Rombach et al. High-Resolution Image Synthesis with Latent Diffusion Models, CVPR’ 22
>
> > **Question 1**: It seems that each 4x4 basic block can represent 2 bits (if B=4), so how to embed 128bit in CIFAR-10? The message should be (28/4)x(28/4)x2bit=98bit?
>
> **Response**: Thanks for your question. The resolution of CIFAR10 and CIFAR100 [1] is 32x32. Thus, the message length for CIFAR should be (32/4)x(32/4)x2bit=128bit. We updated this calculation process in the Appendix C.2 for readers’ convenience.
>
> [1] Krizhevsky et al. Learning multiple layers of features from tiny images. (2009): 7.

---

> > ### Comment · Reviewer_Bfvt · 2023-11-21
> > **Thanks for your response.**
> >
> > Your rebuttal cannot convince me.
> > 1. "During detection, we resize all the data back to 32x32 before inputting them to the detector." I wonder how to get the resize factor after attack? The way you provided is not practical.
> > 2. The resolutions of Labelme and OpenImage datasets used in [1] is much higher than 512x512

---

> > > ### Author Response · Authors · 2023-11-22
> > > **Response to reviewer Bfvt**
> > >
> > > Thanks for providing your insightful feedback.
> > >
> > > We attempt to address your **first concern about resize** by answering the following questions:
> > > - *Why is it necessary to resize it back to 32x32?*
> > >
> > >     The necessity of resizing images to their original size stems from the fixed sizes of the input requirements of the watermark decoder during both the protection and audit phases. Maintaining a consistent image size, identical to that of the initially released protected data, is crucial for the detector's accurate functioning. It's noted that this isn't unique to our approach; it's a standard practice across all baseline methods, which also use watermark decoders with predetermined input sizes. For example, DFD and HiDDeN both take a fixed size of input and resize the images to this fixed size before detection.
> > > - *Why is it still practical?*
> > >
> > >     In practical scenarios, image protectors who own the protected images are aware of the original size of the protected images. Therefore, in cases where they come across images suspected of being generated by a GDM using copyrighted material, they can conveniently resize these images to match the known size of the original protected data, enabling the effective application of the watermark decoding process. The practicality of this setting stems from the protector's knowledge of the original size of their protected images and the ease to restore the original size of the images for effective watermark detection.
> > > - *Whether this will impact the performance?*
> > >
> > >     From Table 12 we can see that no matter the generated images are resized to a larger or a smaller size, as long as the resized images can be resized back to their original size by the images protector, the bit accuracy of the detection can maintain very good performance. This performance, in terms of bit accuracy, is significantly superior to the two baseline methods, HiDDeN and DFD.
> > >
> > > [*Table 12*]
> > > ||DFD|HiDDeN|Ours|
> > > |-|-|-|-|
> > > |No corruption|93.57%|98.93%|99.99%|
> > > |Resize (Small)|92.38%|83.13%|99.30%|
> > > |Resize (Large)|93.20%|79.69%|99.99%|
> > > |Watermark Removal [1]|91.11%|82.20%|99.95%|
> > > |Gaussian Noise|68.63%|83.59%|81.93%|
> > > |Low-pass filter|88.94%|81.05%|99.86%|
> > > |Greyscale|50.82%|97.81%|99.81%|
> > > |JPEG Comp.|62.52%|74.85%|94.45%|
> > >
> > > Regarding the performance with **high-resolution images**,  we hope that our answers for the following questions can address your concerns.
> > > - *Why do we choose these resolutions?*
> > >
> > >     Although some watermarking techniques consider protecting images with high resolutions up to 1024x2048, we note that the image size considered by existing diffusion models papers is usually lower than 512x512. For example, the default size of Stable Diffusion [3], which is used by us in the experiments in Section 4.4, is 512x512. And the default size of DDPM is typically lower than or equal to 256x256 [1][2].
> > >
> > >     [1] Ho et al. Denoising Diffusion Probabilistic Models, NeurIPS’ 20
> > >
> > >     [2] Nichol et al. Improved Denoising Diffusion Probabilistic Models, ICML’ 21
> > >
> > >     [3] Rombach et al. High-Resolution Image Synthesis with Latent Diffusion Models CVPR’ 22
> > > - *What will happen if we increase the resolution?*
> > >
> > >     In response to your concerns, we have also conducted the experiments with resolution higher than 512x512. However, due to the high computational costs of the training of the diffusion models on high resolution images, it takes much longer time to get the results. We will update the results in the revision once we get them.
> > >
> > >     Based on the previous results on images with size from 32x32 to 256x256, we can deduce certain trends regarding performance changes as image resolution increases. We summarize the trend of the change of the bit accuracy when the resolution is increased in the following table. From the table we can see that when the size of the images is increased, there is no obvious deduction in the bit accuracy and our method can consistently achieve near-perfect performance. Therefore, although we haven’t got the results for the experiments with higher resolutions, we believe that following this trend, when the image resolution continues to increase, the general performance of our methods can still be satisfactory.
> > >
> > > |Image size|32x32(CIFAR-10)|64x64(STL-10)|256x256(ImageNet)|
> > > |-|-|-|-|
> > > |Bit acc. of ours(4/255)|100.0000%|99.8883%|99.9524%|

---

> > > > ### Author Response · Authors · 2023-11-22
> > > > **Looking forward to your reply**
> > > >
> > > > Thanks again for your feedback. We hope that our answers have addressed your concerns. If you have any further concerns, please let us know. We are pleasant to address all your concerns during the discussion period.

---

> > > > > ### Author Response · Authors · 2023-11-23
> > > > > **A friendly reminder**
> > > > >
> > > > > We appreciate your reviews. We hope that our responses have adequately addressed your concerns. Should you have any additional questions or points you'd like to discuss, please feel free to let us know. We are keen to engage in further discussion.

---

> ### Author Response · Authors · 2023-11-17
> **A friendly reminder**
>
> We are grateful for the useful comments provided by you. We hope that our answers have addressed your concerns. If you have any further concerns, please let us know. We are looking forward to hearing from you.

---

> ### Comment · Reviewer_Bfvt · 2023-11-23
> **Thanks for your reply (2)**
>
> In real world, an image will under many geometric attacks. I notice that you claime "image protectors who own the protected images are aware of the original size of the protected images".
> First of all I don't agree with you on this, it is very resource consuming to store his original information for every image. Secondly, even if the owner of the image knows the original size, what if the image has been cropped?
>
> Therefore, your reply still cannot convince me. Based on your inadequate thinking, I'm going to decrease my score.

---

> > ### Author Response · Authors · 2023-11-23
> > **Response to reviewer Bfvt**
> >
> > Thanks for your feedback. Please find our response below.
> >
> > **Regarding the concern related to storing information for resizing:**
> >
> > Firstly, we agree that storing the original details for every single image is not practical. However, this step is not necessary since in our scenario, we assume that the released images will adhere to one or a few predetermined sizes. This assumption is reasonable since  in reality, the image owners can control the size of their images before the images are released. In this way, in the audit stages, we can try resizing the images to the predetermined size and directly check the bit accuracy on the image. Secondly, we note that all contemporary watermarking techniques (like the baseline methods DFD[1] and HiDDeN[2]) rely on a consistent image size for effective decoding. Alterations in size can disrupt the ability to extract copyright information embedded in the watermark. Resizing images back to their original or predetermined size is a common practice for watermarking systems in general.  Actually our experimental results also validate that these watermark techniques are robust to this resizing. Therefore, resizing the image back to the original size is a commonly adopted and practical setting in the watermarking process.
> >
> > **Regarding the concern related to the cropping and resizing:**
> >
> > a) If the image is heavily cropped, there is no serious concern about the copyright issue because the cropped part preserves little information of the original image.
> > b) If the image is slightly cropped, the performance of all watermarked methods (like the baseline methods DFD[1] and HiDDeN[2]) may decrease since they all use a detector with a fixed size. Based on our initial experiments, they still can achieve around 80% accuracy when slightly cropped.  Meanwhile, we can easily enhance our method to this problem. If the images are cropped and resized, the blocks are also cropped and resized which is the reason for potential performance drop. Therefore, we can add crop-resize augmentation to train our block detectors, which can further increase the robustness to cropping.
> >
> > We hope this can clarify your concern and we are pleased to have further discussion if you have more concerns.
> >
> > [1] Yu et. al Artificial Fingerprinting for Generative Models: Rooting Deepfake Attribution in Training Data.
> >
> > [2] Zhu et. al HiDDeN: Hiding Data With Deep Networks

---

> ### Comment · Reviewer_Bfvt · 2023-11-23
> **Thanks for your reply (3)**
>
> About the response **Regarding the concern related to storing information for resizing:**
> - I don't agree "resizing the image back to the original size is a **commonly** adopted and practical setting in the watermarking process"
> - I don't agree "we note that all **contemporary watermarking techniques** (like the baseline methods DFD[1] and HiDDeN[2]) rely on a consistent image size for effective decoding".
>
> I hardly see any recent work that would require an advance resizing operation in the face of a resize attack. If you continue to insist, please provide the corresponding reference.
>
> About the response **Regarding the concern related to the cropping and resizing:**
> - How would you define a slight crop? Please provide a specific value for cropping as described in [1].
> - Please focus on more recent work as we are now approaching the end of the year 2023.  "the performance of **all** watermarked methods (like the baseline methods DFD[1] and HiDDeN[2]) may decrease since they all use a detector with a fixed size"
> -  "If the images are cropped and resized, the blocks are also cropped and resized which is the reason for potential performance drop. Therefore, we can add crop-resize augmentation to train our block detectors, which can further increase the robustness to cropping." Please provide experimental results to support your claim.
>
>
> [1] Guo et al. practical deep dispersed watermarking with synchronization and fusion, ACM MM'23

---

> > ### Author Response · Authors · 2023-11-23
> > **Response to reviewer Bfvt**
> >
> > **Regarding the concern related to resizing:**
> > > I hardly see any recent work that would require an advance resizing operation in the face of a resize attack. If you continue to insist, please provide the corresponding reference.
> >
> > **Response:** In response to your requirement, we have provided the specific setting of the resizing-related experiments considered in other recent watermarking papers as follows.
> >
> > MBRS [2], CIN [3], HiDDeN [5]: When considering the potential resizing attack, those papers consider the scenario that the target size of the resizing attack is known by the data protector. Therefore they are able to directly modify their watermark decoder to adapt to the target image size. We note that this assumption is even stricter than ours because in real practice, the image protector should not control what size the image will be transferred to when facing potential attack.
> >
> > DFD [4], Zhao et al. [6]: did not consider defending against resizing and used a detector with fixed size
> >
> > For [1] suggested by the reviewer, it indeed does not require an advance resizing operation but the work is released on 10/23/2023 which is almost one month later than the ICLR deadline.  Meanwhile please note that [1] is not designed specifically for diffusion models.  We will add the discussion on [1] in our revision.
> >
> > **Regarding the concern related to cropping:**
> >
> > > How would you define a slight crop? Please provide a specific value for cropping as described in [1].
> >
> > **Response:** We define that a slight crop as “Randomly crop a a𝐻 × b𝑊 region from the watermarked image, a, b=0.7” We use this crop value and follow the setting of the “crop” experiments of the papers [2][3][5] to test the performance of our method against cropping. The results show that even without considering cropping as an augmentation in the training of the watermark, our method can still achieve bit accuracy of 82.48%.
> >
> > > Please focus on more recent work as we are now approaching the end of the year 2023. "the performance of all watermarked methods (like the baseline methods DFD[1] and HiDDeN[2]) may decrease since they all use a detector with a fixed size"
> >
> > **Response:** As mentioned above, many recent works [2][3][4][5][6] have a fixed size for their detector except the one [1] suggested by the reviewer which is the most recent one and is released later than the ICLR deadline.
> >
> > > If the images are cropped and resized, the blocks are also cropped and resized which is the reason for potential performance drop. Therefore, we can add crop-resize augmentation to train our block detectors, which can further increase the robustness to cropping. Please provide experimental results to support your claim.
> >
> > **Response:** In response to your requirement, we are conducting the experiments. Although we haven’t got the results yet, we believe that our method can also achieve good performance given its outstanding performance in other types of attacks. We will update the results in the revision once we get them.
> >
> > [1] Guo et al. practical deep dispersed watermarking with synchronization and fusion, ACM MM'23
> >
> > [2] Jia et al. Mbrs: Enhancing robustness of dnn-based watermarking by mini-batch of real and simulated jpeg compression, ACM MM 2021
> >
> > [3] Ma et al. Towards Blind Watermarking: Combining Invertible and Non-invertible Mechanisms, ACM MM 2022
> >
> > [4] Yu et. al Artificial Fingerprinting for Generative Models: Rooting Deepfake Attribution in Training Data. 2022
> >
> > [5] Zhu et. al HiDDeN: Hiding Data With Deep Networks. 2018
> >
> > [6] Zhao et al.  A Recipe for Watermarking Diffusion Models. 2023

---

> > > ### Comment · Reviewer_Bfvt · 2023-11-23
> > > **Thanks for your reply (4)**
> > >
> > > Before you get the results, could you answer some questions in case I misunderstand your methodology.
> > > - Take CIFAR as an exmple, it has (32/4)x(32/4)=64 blocks to embed watermark and you set the bit length is 128bit. If leftmost 32x4 region is cropped, that leaves only (28/4)x(32/4)*2=112bit. In this case, how to decode 128bit?
> > > - What the role of the proposed detector in your paper.  Detect if an image has a watermark? If right, why do you need 10 classes (mentioned in Appendix C)?

---

> > > > ### Author Response · Authors · 2023-11-23
> > > >
> > > > Thanks for your feedback.
> > > >
> > > > **To response your first question,** we answer from two settings:
> > > > - Crop and refill with 0-valua pixels. (Settings of [1][2][3])
> > > >     Following the settings of crop experiment in [1][2][3], i.e., the cropped region is refilled with 0-value pixels, our method can still achieve bit accuracy of 82.48% when “Randomly crop an a𝐻 × b𝑊 region from the watermarked image, a, b=0.7”. We can detect in this case. We believe just refilling the pixels by 0 are enough to handle the case you proposed in your first question but we discuss a harder setting below.
> > > >
> > > > - Crop and resize to the fixed size of detector (A even harder setting)
> > > >     We are also running the experiment to resize to original size instead of refilling with 0-value pixels. We use augmentation of resize to train the detector and to increase the robustness for cropping. However, we **directly test that even without augmentation**, when 10% of one edge is cropped and resize, we can achieve bit accuracy of 66.09%. This is significantly higher than non-watermarked samples. If we use AUROC to evaluate it, AUROC of our method is 1, which means the watermark sample can be almost perfectly distinguished from the non-watermarked samples.
> > > >
> > > > Combining the two settings above, we believe it is enough to show our effectiveness under crop without necessity for the last result of training the detector with augmentation. We hope this can clarify your concern. We will add the final result to the updated revision.
> > > >
> > > >
> > > > **To response your second question:**
> > > >
> > > > - Regarding your confusion related to the use of "watermark detector" in Appendix C.1., we hope to make it clear by the following answer:
> > > >
> > > >    Appendix C.1 is not the setting for our main result of DiffusionShield. It is for the preliminary study (Section 3.2) to show the effectiveness of Pattern Unifomity. The watermarks we use in Appendix C.1 ,i.e. the preliminary study, are different from DiffusionShield. We introduce the setting of preliminary study in the followings again. In preliminary study, we use images (pixel values rescaled to [0, 8/255]) as watermarks. These images are selected from 10 classes. To train the detector, we use images from 10 classes, but to train the DDPM, we use 5 classes. In this way, the generated images only are possible to contain 5 classes. Then we use detector to test, if the detector classifies the watermark are from the 5 classes, then we think the watermarks are learned by DDPM. Please refer more details in Section 3.2 and Appendix C.1. And we also show explain our setting for DiffusionShield below again.
> > > >
> > > > - What is the role of watermark detector of DiffusionShield?
> > > >
> > > >    The role of the watermark detector is to decode the message embeded in the watermarks which are added on the images. Based on our design, if there is no watermark by our methods on the images, the decoded message by the decoder will be just "000000..."
> > > >
> > > > - Why we consider 10 classes in DiffusionShield?
> > > >
> > > >     We use 10 classes and only watermark one of the 10 classes to simulate a more realistic but even harder scenario. If all the samples are watermarked, the watermark rate is high and the watermarks are easy to reproduce in the generated samples. However, we only watermark part of samples, i.e one class of the 10 classes, to make the setting more realistic but even harder. Because the unauthorized data offender may use Generative Diffusion Models to train on both watermarked samples and non-watermark samples. The watermark samples are those we protected, while the non-watermarked samples are collected from other resources that are different from our protected data and not related to our copyright. In this harder setting, our method can still perform very well.
> > > >
> > > > We hope this can clarify your concern and we are always pleased to have further discussion if you have more concerns.
> > > >
> > > > [1] Jia et al. Mbrs: Enhancing robustness of dnn-based watermarking by mini-batch of real and simulated jpeg compression, ACM MM 2021
> > > >
> > > > [2] Ma et al. Towards Blind Watermarking: Combining Invertible and Non-invertible Mechanisms, ACM MM 2022
> > > >
> > > > [3] Zhu et. al HiDDeN: Hiding Data With Deep Networks. 2018

---

> ### Comment · Reviewer_Bfvt · 2023-11-23
> **Thanks for your reply (5)**
>
> I'm sorry that I'am still confuse about your reply. Following is my question:
>
> **Further discussion about the first question:**
>
> "If leftmost 32x4 region is cropped, that leaves only (28/4)x(32/4)*2=112bit. In this case, how to decode 128bit?" Could answner this question? In this case, can only 112bits be extracted?
>
>
> **Further discussion about the second question:**
>
> - The purpose of the watermark detector is to decode the watermark, not to detect whether the image has a watermark, right? If so, I suggest to replace "detector" with "decoder", which will be more clarity.
> - I still confuse about the 10 classes. If the decoded bit is "1", what the result of the model? and If the decoded bit is "0", what the result of the model? and If the image is without watermark, what the result of the model? Could you provide an example?
> - How to understand "and then is classified into $b_{1,m}$ in a patch-by-patch manner." (in page 5). Does it mean the input size of your watermark detector is $8u\times8v$, and the output size of your watermark detector is 64bit? Or does it mean the input size of your watermark detector is $u\times v$, and the output size of your watermark detector is 1bit, and you need to decode 64 $u\times
>  v$ blocks to get 64bit?

---

### Official Review · Reviewer_SzGB · 2023-11-09

**Soundness:** 2 fair
**Presentation:** 4 excellent
**Contribution:** 4 excellent
**Rating:** 6
**Confidence:** 5

**Summary:**

The paper introduces an image watermarking technique designed to protect data from being employed in the training of diffusion models. This problem is of significant importance, and the current literature lacks such watermarking solutions. The authors present findings indicating that the "pattern uniformity" metric significantly influences diffusion models' capacity to replicate the watermark. They demonstrate that existing watermarking approaches exhibit inadequate pattern uniformity, rendering them inappropriate for this task. Empirical results in the paper showcase the superior performance of DiffusionShield in comparison to existing methods, even when working with a more limited perturbation budget.

**Strengths:**

+ The paper offers a comprehensive examination of how "pattern uniformity" impacts the watermark's effectiveness in the specific task at hand.

+ In the absence of any corruption, the suggested watermark achieves a high level of bit accuracy while requiring a relatively low perturbation budget.

**Weaknesses:**

+ Insufficient evidence exists to establish the reliability of the classifier that is used to filter the generated images that require protection.

+ The paper does not include an examination of the method's false positive rate and AUROC.

+ The paper does not offer an analysis of the robustness of their watermark against recent attacks known to compromise imperceptible watermarks.

+ Employing the same watermarking pattern for every image simplifies the learning process for potential attackers, making it easier to potentially break the watermark.

**Questions:**

The class-conditional evaluation lacks practicality, as real-world offenders typically do not assign specific class labels to the protected training data used for training diffusion models. In the case of unconditional generation, no definitive ground truth criteria are established for identifying which generated samples should be considered as protected. Currently, the authors define these protected images as those filtered using a classifier, which we will refer to as C. This approach raises concerns as the authors have not sufficiently demonstrated the quality of classifier C, and it may not accurately gauge the impact of protected training data on image generation. A more effective approach to assigning ground truth labels to generated images might involve utilizing an influence function to quantify the influence of protected training data on image generation.

It is imperative to conduct an analysis of the method's false positive rate, specifically to determine if the method assigns high bit accuracy to samples that should not be considered as protected data. This assessment could be performed using metrics like AUROC.

Regarding Table 4, the method exhibits a relatively low resistance to Gaussian noise. It is essential for the authors to specify the standard deviation (std) of the Gaussian noise applied in this context. Recent research has demonstrated the vulnerability of imperceptible watermarks to attacks involving significant Gaussian noise addition and subsequent denoising using diffusion models [1] [2]. Given that DiffusionShield relies on an imperceptible watermark, the authors should provide a thorough analysis of the watermark's robustness against such attacks, including assessments of bit accuracy and AUROC.



[1] Robustness of AI-Image Detectors: Fundamental Limits and Practical Attacks. Saberi et. al., 2023

[2] Invisible Image Watermarks Are Provably Removable Using Generative AI. Zhao et. al., 2023

---

> ### Author Response · Authors · 2023-11-16
> **Response to reviewer SzGB (1/2)**
>
> Thanks for your insightful and detailed comments. We hope the following responses can help address your concerns.
>
> > **Weakness 1 & Question 1**: The protection scenario involves considering a class-conditional model or using a classifier to filter the generated images that require protection in unconditional generation, which may be not practical. The effectiveness of the classifier is not demonstrated.
>
> **Response**: Thanks for your question. We want to clarify that both the class-condition and the classifier in the unconditional case are only used for experimental evaluation. We use them to test the performance of our watermark but do not rely on the classifier or class-condition in real-world scenarios. Below we discuss two viable examples of using the watermark to reveal the infringement in practice:
>
> - First example: Assuming that the unconditional model by data offender generates watermarked and unwatermarked images and the offender releases them. To find the infringement, the data owner does not need to first find the images that are similar to their artworks before watermark detection. They can just use the watermark detector to detect from all the images (both watermark-protected and unwatermarked). Then the watermarked images can be effectively selected. The data owner can choose to engage some human labor to further make sure that the watermarked images are copied from their owned artworks like similar art style and similar object design. Both the watermark and the copied similar things can work as the evidence to reveal and accuse the infringement.
> - Second example: The data owner notices some generated images are very similar to their artwork. They doubt that the images come from generated models trained with their protected watermarked protected images. They use the watermark detector to detect the watermarks in the suspect images. If the watermark is detected, this provides the evidence to reveal and accuse the infringement and then protect their copyright.
>
> Besides, we also demonstrate the effectiveness of our classifier in evaluation to show the correctness of our experimental evaluation. We show TPR (True Positive Rate) and FPR (False Positive Rate) of the classifiers in the following Table 9. Here, TPR reflects the classifier's accuracy in correctly labeling generated images from protected classes as protected. Conversely, FPR indicates the proportion of generated images that, although not from protected classes, were mistakenly identified as protected by the classifier. We can see that generally the FPR is very low, meaning that the bit accuracy of unconditionally generated images is mainly measured on the images from the protected classes, thus the results related to the unconditional model are valid.
>
> In summary, in practical applications, we do not need such classifiers and we only use them for experimental evaluation. The demonstrated ability of classifiers shows that our experimental results are valid. We hope this can clarify your concern about practicality.
>
> [*Table 9*]
> |Dataset|TPR|FPR|
> |-|-|-|
> |CIFAR-10|93.500%|0.667%|
> |CIFAR-100|93.000%|0.061%|
> |STL-10|89.600%|1.356%|
>
> > **Weakness 2 & Question 2**: The paper does not include an examination of the method's false positive rate and AUROC.
>
> **Response**: Thank you for the suggestion to include FPR and AUROC in our watermark detection accuracy assessment. In the following Table 10, we show the TPR, FPR and AUROC of our DiffusionShield with CIFAR-10 dataset. The threshold for the TPR and FPR is 90%, which means that, if more than or equal to 90% of the detected message match the message encoded to the watermarks on the training data, then the image will be considered as an image generated by a model which was trained on the protected data. From the table we can see that our DiffusionShield has a perfect performance in terms of TPR and FPR, which demonstrates DiffusionShield's powerful effectiveness in safeguarding image copyrights.
>
> [*Table 10*]
> |budget=8/255|TPR|FPR|AUROC|
> |-|-|-|-|
> |CIFAR-10|100%|0%|1|
> |CIFAR-100|100%|0%|1|
> |STL-10|100%|0%|0.99|
> |ImageNet|100%|0%|1|

---

> ### Author Response · Authors · 2023-11-16
> **Response to reviewer SzGB (2/2)**
>
> > **Weakness 3 & Question 3**: Provide an analysis on the robustness against recent attacks known to compromise imperceptible watermarks [1][2] & More details about the Gaussian Noise added to the images in the experiments in Table 4.
>
> **Response**: Thank you for your suggestions on the analysis of the robustness of watermarks. In Table 11 we demonstrate the bit accuracy and AUROC of our DiffusionShield together with two baselines methods under the context of watermark removal attempts by [2]. The results show that DiffusionShield is able to survive and achieve a high bit accuracy and AUROC in the experiments on CIFAR-10 and Pokemon. In comparison, the performance of HiDDeN is much worse. Although generally DFD performs well in defending against the removal method, we note that it requires a much higher budget compared to our method, and larger budgets are more difficult to remove.
> Regarding the experiments in Table 4, the standard deviation of the Gaussian Noise is 0.1 and its effect on the images can be visualized in Figure 13 in Appendix J. From the figure we can see that the Gaussian noise considered in Table 4 has a great influence on the quality of images. Even with such an influence on the images, DiffusionShield can still achieve bit accuracy higher than 80%, and AUROC=1, demonstrating the robustness of DiffusionShield against Gaussian Noise.
>
> [*Table 11*]
> ||DFD||HiDDeN||Ours (8/255)||
> |-|-|-|-|-|-|-|
> ||Bit Acc.|AUROC|Bit Acc.|AUROC|Bit Acc.|AUROC|
> |CIFAR-10|91.11%|1.000|82.20%|0.972|99.95%|1.000|
> |ImageNet|79.43%|1.000|48.53%|0.192|54.30%|0.500|
> |pokemon|77.83%|0.958|60.73%|0.970|86.67%|1.000|
>
> [1] Robustness of AI-Image Detectors: Fundamental Limits and Practical Attacks. Saberi et. al., 2023
> [2] Invisible Image Watermarks Are Provably Removable Using Generative AI. Zhao et. al., 2023
>
> > **Weakness 4**: Employing the same watermarking pattern for every image simplifies the learning process for potential attackers, making it easier to potentially break the watermark.
>
> **Response**: Thank you for your helpful observation and question. Although our watermarks share the same patterns for each image, given that the budgets for the watermarks are relatively low, it is still hard for the potential attackers to detect the existence of the watermarks. Even in hidden representation, it is still hard to detect the difference of watermarked images. In Figure 15 in Appendix J, we illustrate and compare the feature representations of a set of unaltered images from one class with their corresponding watermarked versions, along with images from a different class.  The feature extractor is from a Resnet18 Model pre-trained by Contrastive Learning. From the figure we can see that the watermark of DiffusionShield makes almost no difference to the features of images, demonstrating that it is not easy for the potential attacker to discover the watermark in feature space and search for a way to break it. Besides, based on our study in watermark removal attack, even though the offender knows there is watermark, it is hard to remove them, which provides a robust protection for the copyright of the artworks.

---

> ### Author Response · Authors · 2023-11-17
> **A friendly reminder**
>
> We are grateful for the useful comments provided by you. We hope that our answers have addressed your concerns. If you have any further concerns, please let us know. We are looking forward to hearing from you.

---

> > ### Author Response · Authors · 2023-11-22
> > **Looking forward to your reply**
> >
> > Thanks again for your feedback. We hope that our answers have addressed your concerns. If you have any further concerns, please let us know. We are pleasant to address all your concerns during the discussion period.

---

> > > ### Comment · Reviewer_SzGB · 2023-11-22
> > >
> > > I would like to thank the authors for their responses.
> > >
> > > > In summary, in practical applications, we do not need such classifiers and we only use them for experimental evaluation.
> > >
> > > My concern was about the evaluation of your watermark and not its practicality. However, I understand that assigning ground truth labels to generated images is not a simple task, and hence, the authors had to use a classifier. This work would have been more comprehensive if it included more evaluation frameworks.
> > >
> > > >  In Table 11 we demonstrate the bit accuracy and AUROC of our DiffusionShield together with two baselines methods under the context of watermark removal attempts by [2].
> > >
> > > The difference between the performance of DiffusionShield on different datasets is notable. Assuming that the attack is implemented correctly, the authors must add an explanation about this difference, especially since ImageNet is a more diverse and practical dataset compared to the other two.
> > >
> > > In general, I believe that imperceptible watermarks such as DiffusionShield cannot be robust to existing and future attacks, and hence, the watermark proposed by this work is not reliable to be used in practice. However, the analysis of this paper on the requirements of a watermark to be utilized for image copyright protection (i.e., pattern uniformity) is insightful and novel.
> > >
> > > I would like to raise my score to 6 for the authors' efforts in addressing my inquiries and concerns. I expect the authors to add more discussion and explanation on the robustness of their method against attacks such as diffusion purification in their final draft.

---

> > > > ### Author Response · Authors · 2023-11-23
> > > >
> > > > Thank you for your valuable feedback. We appreciate your positive comments on the analysis of pattern uniformity and your suggestions on the evaluation framework and the robustness. We would like to have more discussion about the robustness in the final draft and will improve the evaluation method in our future work. Thanks again for your comments.

---

### Meta-Review · Area_Chair_Phbg · 2023-12-07

**Metareview:**

The paper focuses on the robustness of watermarking in generative diffusion models (GDMs), aiming to protect copyright in the context of generated images. While the paper presents a novel approach and includes a detailed analysis, the reviewers have raised several significant concerns. These include insufficient evidence of the classifier's reliability, lack of examination of false positive rates and AUROC, the method's vulnerability to common attacks, and the practicality of the experimental settings. The reviewers also pointed out the need for a more thorough security analysis against known attacks and questioned the method's robustness against image resizing and cropping. The authors have attempted to address these concerns in their response but have not fully convinced the reviewers.

**Justification For Why Not Higher Score:**

the paper has shortcomings in experimental design and robustness analysis. The method's reliance on watermarking patterns that are uniform across all images poses a significant security risk, making it easier for attackers to learn and potentially break the watermark. The lack of comprehensive tests against common attacks, such as JPEG compression and resizing, limits the paper's practical applicability. Furthermore, the method's effectiveness in handling high-resolution images, which are typical in generative models, remains unverified. The paper also falls short in adequately addressing concerns about practical scenarios, such as image cropping and resizing.

**Justification For Why Not Lower Score:**

NA

---

### Decision · Program_Chairs · 2024-01-16

Reject